# The Effects of Fishmeal Replacement with Degossypolled Cottonseed Protein on Growth, Serum Biochemistry, Endocrine Responses, Lipid Metabolism, and Antioxidant and Immune Responses in Black Carp (*Mylopharyngodon piceus*)

**DOI:** 10.3390/ani15101404

**Published:** 2025-05-13

**Authors:** Jiao Wei, Yifan Fu, Shinan Feng, Jinjing Zhang, Yuanyuan Zhang, Jiaxing Yu, Pengtian Kang, Chenglong Wu, Haifeng Mi

**Affiliations:** 1National-Local Joint Engineering Laboratory of Aquatic Animal Genetic Breeding and Nutrition (Zhejiang), Huzhou University, 759 East 2nd Road, Huzhou 313000, China; weijiao0323@outlook.com (J.W.); fuyifan1210@outlook.com (Y.F.); xfnn18@outlook.com (S.F.); zjj18388550747@outlook.com (J.Z.); y2424507372@outlook.com (Y.Z.); yujiaxing313@outlook.com (J.Y.); 2Gansu Provincial Aquatic Technology Extension Station, 113 Zhongshan Road, Lanzhou 730030, China; 3Healthy Aquaculture Key Laboratory of Sichuan Province, Tongwei Co., Ltd., 588 Tianfu Avenue, Chengdu 610093, China; mihf@tongwei.com

**Keywords:** *Mylopharyngodon piceus*, degossypolled cottonseed protein, biochemical indices, antioxidant capacity, immune responses

## Abstract

Although fishmeal is an important protein source in aquafeeds, its high cost has led to the search for alternative replacements. This study aimed to investigate the effects of replacing fishmeal with degossypolled cottonseed protein (DCP) on growth, serum biochemistry, endocrine responses, lipid metabolism, antioxidant activity, and immune responses in black carp. The results indicated that an appropriate level of fishmeal replacement with DCP (20%) could improve serum biochemical parameters, digestive activity, antioxidant capacity, and immunity in black carp. These findings provide a theoretical reference for applying DCP in black carp feed.

## 1. Introduction

Fishmeal (FM) is an important dietary ingredient and has many advantages due to its high protein content, balanced amino acid composition, good palatability, and high digestibility [1]. However, due to its decreased production worldwide, the price of FM has risen sharply in recent years, which has affected the sustainable and rapid development of modern aquaculture [2]. Therefore, as a research hotspot, the search for suitable FM substitutes has focused on animal and plant protein resources in recent years [1,3,4]. Compared with animal protein resources with higher market prices, many plant protein resources are more widely available, cheaper, and have high nutritional value, with high contents of protein, lipids, and vitamins as nonfood protein resources [2]. A number of research studies have been carried out on FM substitution with different plant protein resources, including cottonseed meal [5], fermented soybean meal [6], and peanut meal [7]. These studies have shown that appropriate levels of plant protein resources replacing FM in the feed did not affect normal growth in aquatic animals such as golden pompano (*Trachinotusovatus*) [5], largemouth bass (*Micropterus salmoides*) [6], and hybrid grouper (*Epinephelus fuscoguttatus*♀ × *Epinephelus lanceolatus*♂) [7].

As an important plant protein with a wide range of sources and a low price, cottonseed meal is the main by-product after cottonseed processing. It is rich in protein, and it is an ideal plant protein source [8]. Meanwhile, cottonseed meal contains antinutritional factors such as free gossypol and tannin, which can adversely influence the health and growth performance of animals [9,10,11]. In order to reduce these adverse factors, a low-temperature and two-step solvent-extraction process has been developed and used to produce high-quality degossypolled cottonseed protein (DCP) that removes free gossypol, aflatoxins, and water-soluble antinutritional factors and protects the quality of crude protein and amino acids [12,13]. Processing is mainly used to reduce the content of water-soluble non-starch polysaccharides, oligo-oligosaccharides, and antinutritional factors (cotton-phenol and tannins), etc. [14]. Recently, many studies have been conducted on the application of cottonseed and its products in aquatic feed, such as rainbow trout (*Oncorhynchus mykiss*) [12], juvenile golden pompano (*Trachinotus ovatus*) [15], pearl gentian grouper (*Epinephelus fuscoguttatus*♀ × ♂*Epinephelus lanceolatu*♂) [16], and Japanese seabass (*Lateolabrax japonicas*) [17]. However, these studies were mainly focused on growth, body composition, hematological indices, antioxidant capacity, and innate immunity; there was little comprehensive information that detailed the impacts of FM’s replacement with DCP in the freshwater carnivorous fish species comprising the traditional “Four Domesticated Fish” in China.

Black carp (*Mylopharyngodon piceus*), a cyprinid fish, is one of four major fish species and is a popular fish in China [18]. According to the China Fishery Statistical Yearbook, the production of black carp reached about 0.80 million tons in 2023, an increase of 6.96% compared with the previous year. Recent studies primarily concentrated on the requirements of carbohydrates [19], selenium yeast [20], leucine [21], different vitamins [22,23], and protein replacement [24]. Previously, our team investigated the effect of replacing FM with blood meal on black carp, a study that ensured the methodological continuity of this experiment [24]. Currently, little research has examined replacing FM with DCP in black carp feeding. Based on these ideas, this experiment looked at growth performance, serum biochemistry, antioxidant capability, inflammatory responses, and muscle quality and analyzed them from the perspective of genes. It aimed to determine the appropriate DCP replacement proportion for FM and provide theoretical support for the formulation of low-FM feed for black carp.

## 2. Materials and Methods

### 2.1. Experiment with Fish Diets

FM, DCP, soybean meal, and corn protein meal were used as protein sources; soy lecithin, rapeseed oil, and fish oil were used as lipid sources; and tapioca flour was used as a carbohydrate source. Next, 0%, 10%, 20%, 30%, 40%, and 50% substitution amounts for FM’s replacement with DCP (DCP0, DCP10, DCP20, DCP30, DCP40, and DCP50), respectively, were configured into six kinds of nitrogen-equivalent and energy-equivalent sinking experiment feeds. DCP was commercially sourced and used the low-temperature and two-step solvent-extraction process, which removed free gossypol, aflatoxins, and water-soluble antinutritional factors. The free gossypol content in DCP was about 150 mg/kg, according to the manufacturer’s detection results. Raw materials underwent sequential processing consisting of crushing, sieving through a 60-mesh screen, precise weighing, and homogeneous mixing. The components with small quantities were mixed using step-by-step expansion processing to produce pellet feeds 2 mm in diameter, which were dried at 35℃ in a hot-air-circulating oven and stored for spare use. The formulations of test feeds for each experimental group and the nutrient compositions are shown in Table 1 and Table 2.

### 2.2. Experimental Procedure

This experiment was in line with the national guidelines regarding the feeding and use of vertebrate animals. It was approved by the Ethical Committee of Huzhou University, located in Huzhou, China. The approval ID is HUZJ-DW-2023-139, and the date of approval was 6 March 2023.

The black carp used in this experiment were purchased from local farmers in Huzhou, Zhejiang, and were placed into a holding tank for a fortnight for domestication and fed with basic diets. After this, 540 black carp—uniform individuals, healthy and disease-free, and with an average body weight of 6.80 ± 0.50 g—were picked and randomly placed in 18 indoor water tanks (500 L). Six groups were set up in this experiment, and three replicates were set up in each group, with 30 fish in each replicate. In order to reduce potential tank effects, all tanks were maintained under identical environmental conditions throughout the experiment. The daily feeding was timed at 08:30 and 17:30. The daily feeding rate was about 3% of the body weight, which was adjusted appropriately according to the feeding situation of the fish and the growth of the fish species. The residual bait was removed after 1 h of baiting, and the daily water exchange was about 1/4–1/3 of the water tanks. During the experimental period, the water temperature was 27.0 ± 0.5 °C, the oxygen was continuously filled for 24 h with the dissolved oxygen above 5.8 mg/L, and the culture cycle was 60 days. The mortality rate was checked and recorded daily to calculate the survival rate.

### 2.3. Sample Collection and Measurement of Growth Performance

At the end of the feeding experiment, all fish in each tank were collected and anesthetized using 300 mg/L of sodium trialkyl methyl-anesthesia-sulfonate (Sigma, St. Louis, MO, USA). The total weight, total length, and quantity of each fish were recorded to evaluate growth performance. Then, using the method described by Wu et al. [21], serum, blood, liver, intestinal, and muscle samples were collected. Fifteen fish were randomly picked from each tank, then blood was collected from the tail vein of these fifteen fish and stored at −80 °C. After collecting and processing, these 15 fish were rapidly dissected on ice, and the visceral mass was dissected and weighed to record their weights. After rapid separation from the intestines, the livers of these 15 fish in each tank were weighed, respectively. The livers, intestines, and muscles were rapidly frozen in liquid nitrogen and then transferred to a −80 °C refrigerator for storage. After sampling, three fish in each tank were selected and stored at −20 °C for the next detection of composition.

### 2.4. Measurement of Fish and Feed Composition

The moisture of the test feeds was determined by drying at 105 °C, while the moisture of the fish body and dorsal muscle was determined by freeze-drying (Alpha2-4 LSC Basic, Martin Christ Gefriertrocknungsanlagen GmbH, Osterode am Harz, Germany); crude protein, crude lipid, and ash amounts were determined using Dumas automatic rapid nitrogen fixation (Rapid N exceed, Elementar Analysensysteme GmbH, Frankfurt, Germany), Soxhlet extraction, and Marfo oven burn (550 °C), respectively.

### 2.5. Measurement of Serum Biochemical Parameters

High-density lipoprotein cholesterol (HDL-C), albumin (ALB), triglyceride (TG), low-density lipoprotein cholesterol (LDL-C), total bile acids (TBAs), and the activities of alanine aminotransferase (ALT), lactate dehydrogenase (LDH), alkaline phosphatase (ALP), glucose (GLU), total cholesterol (TC), and glutamic oxaloacetic acid (AST) were measured with the help of Ningbo Prebis Biotec Co. (Ningbo, China) and detected using kits from Ningbo Prebon Biotechnology Co. (Ningbo, China) Insulin (INS). Insulin growth factor (IGF-1 and IGF-2) and 5-hydroxytryptamine (5-HT) were detected using kits from Shanghai Hengyuan Biotechnology Co. (Shanghai, China). All the analyses contained at least 3 replicates.

### 2.6. Measurement of Physiological and Biochemical Indexes

Lipase (LPS), amylase (α-AMS), trypsin (TRY), and chymotrypsin (CYT) were determined with the help of Jiancheng Biotech (Nanjing, China). The procedure was performed according to the provided instructions. Total antioxidant capacity (T-AOC), glutathione-S-transferase (GST), glutathione peroxidase (GPx), glutathione reductase (GR), catalase (CAT), hydrogen peroxide (H_2_O_2_), superoxide dismutase (T-SOD), reduced glutathione (GSH), and malondialdehyde (MDA) levels in the liver and intestine were measured using kits purchased from Shanghai Hengyuan Biotech (Shanghai, China). Immunoglobulin M (IgM), lysozyme (LZM), and alkaline phosphatase (ALP) kits were purchased from Jiancheng Biotech (Nanjing, China). All the analyses contained at least 3 replicates.

### 2.7. Measurement of Relative Gene Expression

We used Trizol reagent (Invitrogen, Carlsbad, CA, USA) to extract total RNA in liver tissue. The RNA concentration and integrity were evaluated via spectrophotometry and electrophoresis, following protocols described by Song et al. [25]. The experiment employed specific primers targeting molecules involved in digestion, antioxidant systems, immune responses, lipid metabolism, and their respective signaling pathways (Table 3). All primers were synthesized by Biosune Co. (Shanghai, China). β-actin served as the internal reference gene. Quantitative PCR (qPCR) analyses were conducted according to methodologies established by Wu et al. [25] and Zhang et al. [20]. All the analyses contained at least 3 replicates.

### 2.8. Data Processing and Statistical Analysis

The experimental data were expressed as the mean ± standard deviation (mean ± SD). Prior to statistical analysis, the homogeneity of variances was assessed using Levene’s test. The collected data were analyzed using one-way ANOVA in SPSS 26.0. When significant differences were found, multiple comparisons were performed using Tukey’s post hoc test, with a significance level of *p* < 0.05. Tukey’s test was chosen because it is well-suited for comparing all pairwise group differences while controlling for Type I error.

## 3. Results

### 3.1. Growth Performance and Fish Body Composition

The substitution of DCP had no notable effect on the FBW, WG, SGR, FCR, and PER of black carp (*p* > 0.05) (Table 4). Compared with the DCP0 group, VSI, HSI, and ISI showed a significantly upward trend with increasing DCP replacement levels (*p* < 0.05), although no notable changes were observed between the DCP0 and DCP10 test groups (*p* > 0.05). The highest value of CF was obtained in the DCP0 test group (*p* < 0.05). FI in the DCP10 group was markedly higher than that in the DCP30, DCP40, and DCP50 test groups (*p* < 0.05).

The moisture of whole fish displayed a tendency to rise, peaking in the DCP20 group (*p* < 0.05), and then decreasing in the DCP40 group (Table 5). There were no significant differences in the moisture of the dorsal muscle among these six DCP groups (*p* > 0.05). Although the crude protein content of the whole fish body and dorsal muscle decreased with the increase in the substitution level (*p* < 0.05), no distinctions were detected in the whole fish body among the DCP0, DCP10, DCP20, and DCP30 test groups (*p* > 0.05). The crude lipid contents in the whole fish and dorsal muscle increased significantly with the rise in DCP substitution, reaching their highest value in the DCP40 group (*p* < 0.05). Moreover, the ash of the whole fish displayed a decreasing trend (*p* < 0.05), with no notable differences among the DCP0, DCP10, DCP20, and DCP30 groups (*p* > 0.05). Regarding the ash content of the dorsal muscle, no marked differences were observed (*p* > 0.05). Amino acid profiles in dorsal muscle remained unaltered following DCP substitution for FM (Table 6).

### 3.2. Serum Biochemical Parameters

Among the groups, IGF-I showed a significant decreasing trend, reaching the maximum in the DCP30 group and then decreasing to the minimum in the DCP50 group (*p* < 0.05). No marked changes were observed for IGF-I amounts among DCP0, DCP10, DCP20, and DCP30 groups (*p* > 0.05) (Table 7). Similarly, IGF-II levels first increased, peaked in the DCP30 test group (*p* < 0.05), and then decreased in the DCP40 and DCP50 test groups (*p* > 0.05). The INS contents declined as the substitution level rose (*p* < 0.05), and no statistically marked changes were noted among the DCP0, DCP10, and DCP20 test groups (*p* > 0.05). 5-HT revealed a markedly upward trend and peaked in the DCP50 group (*p* < 0.05). No remarkable variations in the HDL-C were displayed among these six DCP test groups (*p* < 0.05). LDL-C concentrations exhibited a first elevation and then a decrease with increasing DCP substitution levels. LDL-C contents showed significant elevation in the DCP40 and DCP50 test groups compared with the DCP0 and DCP10 groups (*p* < 0.05). GLU contents showed a rising trend and reached their peak in DCP40 and DCP50 test groups (*p* < 0.05), although no changes were presented among the DCP0, DCP10, and DCP20 test groups (*p* > 0.05). TG and TC values in the DCP40 test group were notably higher than in other test groups (*p* < 0.05). However, statistical analysis showed no marked differences in the TG and TC contents among the DCP0, DCP10, DCP20, and DCP30 test groups (*p* > 0.05). The lowest AST and ALT values were shown in the DCP30 test group (*p* < 0.05), and their highest values were observed in the DCP10 test group (*p* < 0.05). ALP activities showed an increasing tendency with the level of substitution (*p* < 0.05) and peaked in the DCP40 test group (*p* < 0.05). With the rise in the substitution level, ALB levels exhibited a stable tendency among DCP0, DCP10, and DCP20 test groups (*p* > 0.05) and a decline in the DCP50 group (*p* < 0.05). There was a constantly elevating tendency in serum TBA contents, which notably peaked in the DCP50 test groups (*p* < 0.05). In addition, BUN levels showed an increase first and then maintained a stable trend. BUN levels in the DCP40 and DCP50 test groups were markedly higher than those at the 0% and 10% substitution levels (*p* < 0.05). Moreover, LDH activities first declined and then increased. LDH activities reached a minimum in the DCP30 test group (*p* < 0.05).

### 3.3. Digestive Enzyme Ability and Lipid Metabolism in the Liver and Dorsal Muscle

The α-AMS and TRY activity in the liver both exhibited upward trends, reaching peak values in the DCP40 group, and declined in the DCP50 group (*p* < 0.05) (Table 8). LPS activity showed a significant increasing trend and peaked in the DCP50 test group (*p* < 0.05). In the liver, the expression amounts of α-AMS were notably elevated and then lowered in the DCP50 test group (*p* < 0.05), combined with no differences between the DCP30 and DCP40 test groups (*p* > 0.05) (Figure 1). TRY and BAL transcription levels demonstrated a first upward trend with increasing DCP substitution, reaching the maximum in the DCP40 test group, and then decreased in the DCP50 test group (*p* < 0.05).

High DCP substitution levels notably elevated the expression of fatty acid synthesis genes in the liver of black carp. The expression levels of *acc*, *fas*, *scd*, *fads6*, and *elovl5* were markedly elevated in the DCP40 group (*p* < 0.05) (Figure 2). The 30% substitution level did not affect the expression of *acc* and *elovl5* when compared with the DCP0 group (*p* > 0.05). *Scd* and *fads6* in DCP30 and DCP40 groups displayed a notable elevation compared to the other groups (*p* < 0.05). High DCP diets significantly increased the expression of *fabp* and *gpat3/4* compared with the DCP0 test group (*p* < 0.05). Notably, marked down-regulation levels of *gpat3/4* expression were observed in the DCP50 test group compared with the DCP40 test group (*p* < 0.05). Simultaneously, the expression levels of *plin1/2* exhibited an upward trend in tandem with the increasing substitution level of DCP (*p* < 0.05). However, this tendency was not statistically substantial among the DCP0, DCP10, and DCP30 test groups (*p* > 0.05).

A similar trend in the variation in TG contents was presented in the dorsal muscle and liver samples in black carp. TG contents showed a first increase in tendency with DCP replacement levels, increasing to 40% or 50%. Although no notable changes were observed between the DCP40 and DCP50 groups (*p* > 0.05), their values were significantly higher than those in the DCP0, DCP10, DCP20, and DCP30 groups (*p* < 0.05) (Figure 3).

### 3.4. Antioxidant-Related Parameters

The T-SOD and CAT in the liver exhibited a rising trend as the substitution level increased and then maintained a stable trend (Table 9). Higher activities of T-SOD and CAT were shown in the DCP40 and DCP50 groups compared with the DCP0 group (*p* < 0.05), respectively. However, DCP replacement decreased GPx activity in the hepatic cells of black carp, with no marked differences noticed among DCP0, DCP10, and DCP20 test groups (*p* > 0.05). GSH levels showed a trend of gradual increase, peaked in the DCP30 group (*p* < 0.05), and decreased in the DCP40 and DCP50 groups (*p* > 0.05). No statistical differences in GR activities were detected among the experimental groups (*p* < 0.05). Hepatic GST activities were significantly elevated in response to a 50% replacement ratio (*p* < 0.05). As the substitution level increased, T-AOC values initially decreased and subsequently increased (*p* < 0.05), with the lowest T-AOC value observed in the DCP50 group. H_2_O_2_ contents first declined to the minimum in the DCP30 group and then elevated to the maximum in the DCP50 group. The lowest MDA contents were found in the DCP20 group, while their values were significantly increased in the DCP40 and DCP50 groups (*p* < 0.05).

The expression values of *sod1* were significantly elevated in the DCP10 group compared with the other five DCP groups (*p* < 0.05) (Figure 4). *Sod2* expression values showed a first increase in the DCP20 group and then declined with increasing DCP substitution. The minimum *Sod2* expression values were demonstrated in the DCP50 group compared to the DCP0, DCP10, and DCP20 groups (*p* < 0.05). There were no evident differences in *cat* transcription levels among these six DCP test groups (*p* > 0.05). The *gr* expression level in DCP50 was markedly lower than in the other five groups (*p* < 0.05). *Gst* expression levels showed an initial increase and then a stable trend with elevated DCP substitution levels. Although there were no marked changes among the DCP10, DCP20, DCP30, DCP40, and DCP50 groups, higher *Gst* expression levels were observed in the DCP50 group compared with the DCP0 group (*p* < 0.05). The *gpx* levels in the DCP10, DCP20, and DCP30 groups were notably higher than those in the DCP0, DCP40, and DCP50 groups (*p* < 0.05). *GCLC*, *GCLR*, and *nrf2* were all characterized by notable elevation in the DCP30 group (*p* < 0.05), followed by a decline in the DCP40 group. Although there were no significant variations between the DCP0 and DCP10 test groups (*p* > 0.05), *Keap1* expression levels declined in the DCP20, DCP30, DCP40, and DCP50 groups (*p* < 0.05).

### 3.5. Immune Parameters

LZM contents showed a first marked upward trend, peaked in DCP10 and DCP30 test groups (*p* < 0.05), and then decreased in the DCP50 test group (Table 10). The LZM activities gradually increased and remained stable in the intestine (*p* < 0.05), and no marked differences were observed among the DCP30, DCP40, and DCP50 test groups (*p* > 0.05). ALP activities in the liver and intestine of the DCP20 and DCP30 test groups were both markedly increased compared with the DCP0 and DCP50 test groups (*p* < 0.05). Hepatic IgM levels first increased, peaked in the DCP20 group, and then declined with the rising dietary DCP content (*p* < 0.05). Intestinal IgM levels were notably decreased in the 50% DCP replacement group (*p* < 0.05), although there were no marked changes in IgM levels in the intestine of the other five DCP test groups (*p* > 0.05).

DCP replacement significantly raised the *lzm* expression levels in the liver, with the DCP10 group showing notably higher levels compared to other groups (*p* < 0.05) (Figure 5). The expression levels of *alp* first increased, peaked in the DCP20 groups, and then decreased with the increase in DCP substitution levels (*p* < 0.05). A significant reduction in *igm* expression levels was detected exclusively in the DCP50 group (*p* < 0.05), while no marked variations were detected among the other five groups (*p* > 0.05). The DCP10 group had considerably improved expression levels of *leap-2* compared to the other five groups (*p* < 0.05), and there were no significant changes among the DCP0, DCP20, and DCP30 groups (*p* > 0.05).

## 4. Discussion

Growth performance is a crucial indicator for evaluating the effectiveness of substituting FM with plant protein sources [26]. In the present study, although WG and SGR increased in the DCP30 and DC40 groups compared with the DCP0 group, there were no significant variations in WG and SGR among these six DCP groups; similar results were also shown in juvenile golden pompano [10], rainbow trout [12], and hybrid grouper [7]. Generally, there are tight relationships among normal growth, metabolic ability, and digestive enzyme activities in animals and fish species [27,28]. Greater digestive enzyme activity could improve growth performance by promoting the utilization of exogenous nutrients in fish [27,28]. In our results, higher TRY and LPS activities were also shown in the DCP30 and DCP40 groups compared with the DCP0 group, which is consistent with another study on South American white shrimp [29]. Combined with these findings, this study indicated that a suitable FM substitution level with DCP could maintain or increase normal growth by modulating digestive enzyme activities in black carp. However, previous studies also found that a higher ratio of low-gossypol cottonseed meal could impair growth in silver sillago [30] and juvenile black seabass [31]. Meanwhile, higher contents of cottonseed meal could decrease FI values in Ussuri catfish *Pseudobagrus ussuriensis* [29]; similar results were also presented in our study, which indicated that higher contents of free gossypol could impair growth via lowering animal feed intake in black carp. In addition, higher VSI, HSI, and ISI were shown in DCP40 and DCP50 groups; similar results were observed in channel catfish [32]. A number of studies found that a higher HSI value is always linked with greater hepatic lipid content and weakened lipid metabolic function in cultured fish species [20,22,23]. Combined with greater hepatic TAG contents in the DCP40 and DCP50 groups, it is suggested that higher dietary DCP could impair lipid metabolic function by heightening HSI values in black carp. Moreover, a higher FCR value was shown in the DCP50 group; similar results were also demonstrated in grouper largemouth bass [33,34], golden pompano [10], and red swamp crayfish [35], which may be due to the excessive FM replacement with DCP. This could affect the hardness and palatability of feed and influence the utilization of feed in fish [36,37].

In this experiment, the replacement of FM with DCP did not affect the moisture content of the dorsal muscle of black carp, which is consistent with results in rainbow trout [12]. Previous studies observed that the crude protein of whole fish and dorsal muscle decreased with increasing DCP replacement levels in juvenile largemouth bass [33] and silver sillago [30]; similar results were also shown in our results, which indicated that the DCP diet was less palatable and consumed at a much lower rate than the other diets [30]. Previous studies reported no changes in the amino acid profiles in the fish muscle [31,38]; similar results were also shown in our study, which suggested that greater replacement of FM (50%) with DCP did not influence the amino acid profiles in the muscle of black carp. However, other studies also found that greater FM replacement with a cottonseed protein concentrate (CPC) could reduce the total and essential contents in the muscle of largemouth bass [34] and Amur sturgeon [39]. This diversity may be due to the imbalance of amino acid composition and utilization abilities mediated by the higher dietary DCP content in these fish species.

In addition, the content of crude lipid increased, which is similar to the results found in black sea bass [31]. A previous study also reported that there were higher carbohydrate contents in cottonseed meal as a typical plant-derived protein resource (19.30 ± 0.1%) [40], and higher contents of carbohydrates could induce lipid deposition or synthesis in fish [41]. Combined with the results found for body lipid contents, TAG contents, and HSI values in our results, this indicated that lipid deposition and/or synthesis were induced by higher contents of carbohydrates with increasing dietary DCP replacement levels in black carp. In addition, a former study also found that the significant increase in crude lipid contents may be due to impaired liver function or lower utilization of lipids as an energy source in diets with higher contents of cottonseed meal [42]. However, other studies also found reduced growth and lower body lipid levels in cultured fish fed diets with higher cottonseed meal levels [34,43]; this may be due to the differences in fish species and their utilization abilities corresponding to experimental feed formulations. Although there were no changes in ash contents in the muscle samples, ash contents were reduced in the whole fish body of the DCP40 and DCP50 groups, while there were no differences in whole fish body ash contents in rainbow trout [12], silver sillago [30], and pearl gentian groupers [44], which may be due to the different experimental feed ingredients, culturing conditions, and fish species [45].

Serum parameters of fish are critical indicators to effectively respond to the metabolic ability and health status in cultured fish species [46,47]. Variations in AST and ALT activities could reflect hepatic damage in fish and correlate with metabolic ability [48,49]. In this experiment, the levels of serum AST and ALT were decreased in the DCP30 group, which was consistent with the results in largemouth bass [33] and large yellow croaker [45], indicating that 30% FM replacement with DCP did not induce typical hepatic damage in black carp. Notably, higher AST and ALT levels were observed in DCP0 and DCP10 groups, which may be attributed to metabolic adaptation during transamination and their use for protein synthesis in fish associated with FM and DCP utilization [13]. Regarding increased AST and ALT activities in the DCP50 group, this might be linked to higher levels of residual gossypol present in the DCP inducing metabolic stress at excessive DCP replacement levels [50]. As a metabolite marker, changes in serum GLU could act as a stress marker for evaluating the effects of dietary protein resource replacement [51]. Our results showed that serum GLU contents were heightened with increased levels of DCP replacement, aligning with findings in pearl gentian groupers [44]. Meanwhile, higher activity of α-AMS was demonstrated in the liver and intestine of fish fed diets with higher DCP levels compared with the DCP0 group, suggesting that higher contents of GLU could be generated through the digestion of starch or polysaccharides in DCP and then be transported into the serum [52]. These findings indicate that higher contents of GLU could be induced by higher contents of carbohydrate included in dietary DCP in black carp. Although INS could lower GLU levels, serum INS contents were markedly reduced in these higher levels of DCP replacement, which indicated that higher levels of DCP could impair glucose metabolism by lowering INS secretion, although this regulatory mechanism needed to be further studied in black carp.

Moreover, as key serum hormones, higher levels of IGFs always play critical roles in improving protein synthesis in animals [53]. Further, variations in serum BUN are usually related to nitrogen utilization and impact the protein-synthesizing ability [25,54]. Combined with protein contents in the fish body and muscle and contents of IGF-1/2 and BUN in the serum of DCP40 and DCP50 groups, this indicated that high DCP might weaken dietary nitrogen utilization and reduce protein synthesis by lowering IGF-1/2 contents in black carp. Previous studies reported that 5-HT could act as an inhibitory neuromodulator and decrease food intake in fish [55,56]. In this study, serum 5-HT levels were noticeably heightened in the DCP40 and DCP50 groups in comparison with the DCP0, DCP10, and DCP20 groups. Along with lower FI values in the DCP40 and DCP50 groups, this indicates that FI could be inhibited by greater 5-HT amounts induced by increased dietary DCP, although its regulatory mechanism needs to be well-studied in the future. As the key transporter for lipid molecules out of hepatic cells, LDL-C with higher activities or levels usually indicates cellular damage to hepatic tissue [57,58]. Furthermore, higher TBA contents could be linked to lipid metabolic dysfunction induced by higher starch contents in fish [57]. These increased LDL-C, TG, TC, and TBA amounts in the DCP40 group suggest that high DCP levels could impact lipid metabolism in black carp.

In general, FAS can be used to generate palmitate with cytoplasmic Ac-CoA and Mal-CoA molecules in hepatic cells [59,60]. Palmitate can be used to synthesize corresponding saturated fatty acids (SFAs) and monounsaturated fatty acids (MUFAs), catalyzed by SCD, ELOVLs, FADS, etc. [61]. In line with these previous findings in largemouth bass [60], higher levels of *acc*, *fas*, *fads*, *scd*, and *elovl5* were also shown in the liver of black carp fed higher levels of dietary DCP, which suggests that higher DCP intake could cause fatty acid synthesis by increasing the transcription levels of these genes in the hepatic cells of black carp. FABPs could assist in the transportation of fatty acids to the endoplasmic reticulum for the next generation of TAG within hepatic cells [62]. As rate-limiting enzymes, GPAT3 and GPAT4 play key roles in synthesizing TAG [63]. Meanwhile, as two key coated proteins, Plin1 and Plin2, are tightly linked with the biogenesis and storage of lipid droplets in hepatic cells [64]. Many studies have found that higher expression amounts of Plins could be related to so-called metabolic dysfunction or disorders in animals [65,66,67]. This study also found there were higher levels of *fabp6*, *gpat3*, *gpat4*, *plin1*, and *plin2* in black carp fed higher dietary DCP, which is in line with previous results in largemouth bass [60] and Atlantic salmon [68]. Along with higher TG contents and higher mRNA expression levels of these functional genes, this demonstrates that higher contents of DCP could enhance TAG synthesis in the liver of black carp. Previous research has reported that TAG stored in lipid droplets could be hydrolyzed into FAs and glycerol by adipose triglyceride lipase (ATGL) and lipoprotein lipase (LPL) in animals [69]. Combined with lower *atgl* and *lpl* expression levels in this study, this suggests that higher contents of DCP could inhibit TAG hydrolysis and exacerbate lipid deposition in black carp.

Many studies have found that redox homeostasis always plays a key role in maintaining normal growth and improving immunity in animals [70,71]. These antioxidant enzymes (SOD, CAT, and GPx) can protect cells from oxidative stress. SOD catalyzes the dismutation of superoxide anion radical to O_2_ and H_2_O_2_, CAT catalyzes the decomposition of H_2_O_2_ into water and oxygen, and GPx reduces lipid hydroperoxides to their corresponding alcohols and free hydrogen peroxide to water, thus limiting oxidative damage [72,73]. GST can generate disulfide bonds between GSH molecules, while GR can decompose these disulfide bonds in glutathione disulfide (GSSG), and glutamate–cysteine ligase (GCL) can catalyze the generation of GSH [74,75]. In general, the cellular redox balance is tightly related with variations in these antioxidant enzymes and molecules (SOD, CAT, GPx, GR, GCL, GSH, etc.) [76]. In this experiment, suitable contents of DCP heightened the amounts of T-SOD, T-AOC, GSH, and *gcl*, while higher contents of DCP reduced the *gpx* expression levels in the liver of fish. Meanwhile, DCP50 also elevated the amounts of GST, H_2_O_2_, and MDA in fish liver, which is in agreement with previous results in largemouth bass [33], large yellow croaker [45], and ussuri catfish [13]. Combined with the above results, this indicated that suitable contents of dietary DCP could improve antioxidant capacity, protect cells from ROS-induced damage, and maintain redox balance in black carp [77]. Many studies have proved that the Nrf2/Keap1 signal pathway can regulate redox homeostasis and the expression levels of these antioxidant enzymes in animals [78,79]. Our study found there were higher expression levels of *nrf2* and lower amounts of *keap1* in adequate DCP test groups, which is consistent with previous results found for grass carp [80], common carp [81], and largemouth bass [82]. Combined with these findings, suitable DCP (20%) could promote antioxidant capacities by regulating the Nrf2/Keap1 pathway in black carp.

The detection of immune indices is very important for assessing animal health status and immune regulatory roles mediated by various exogenous nutrients [20,54]. In general, animal immune function could be heightened by increasing the contents of different immune defense molecules, including LZM, ALP, IgM, and LEAP-2 [83,84,85]. LZM and LEAP-2 can act as crucial immune defense effector molecules by destroying the surface structure of these invading pathogens in fish [86]. In the present study, LZM activities and expression levels were both heightened with an adequate amount of DCP in the liver of black carp, which was in line with former results in South American white shrimp [29] and golden pompano [87]. Meanwhile, higher levels of *leap-2* were also observed in fish fed lower DCP diets compared with higher DCP diets; similar results were also shown in fish fed adequate dietary nutrients [23,88]. Thus, higher LZM levels could be induced by adequate dietary DCP and used to improve the immunity of black carp. As a typical phosphohydrolase and diagnostic parameter, ALP can mediate crucial immune defense functions by triggering activities of immune cells and can be used to evaluate health status in fish and other animals [89,90]. Our results also showed there were higher ALP activities and transcription levels in the DCP20 group, which is similar to the result found for grass carp [80]. As a major humoral immune protein produced by B cells, IgM can recognize target epitopes of foreign pathogens and help construct their complex, which is rendered to macrophages and phagocytes to destroy these pathogens in animals [84]. Higher levels of IgM are usually closely related to better humoral immunity in fish fed suitable contents of dietary nutrients [91]. These findings align with previous results found for hybrid grouper [7], where higher levels of IgM were also shown in the liver and intestine of fish fed suitable amounts of DCP. This suggests that adequate dietary DCP could promote humoral immunity, enhance health status, and protect hosts by increasing IgM secretion in black carp. However, lower levels of IgM were also present in fish fed excessive DCP, similar to many cultured fish fed higher contents of DCP or cottonseed meal, respectively [80,92]. These findings suggest that excessive dietary DCP might weaken humoral immunity and health status by reducing the secretion of these defense molecules in black carp.

## 5. Conclusions

In summary, the results of this study showed that a suitable amount of DCP (20%) could improve serum biochemical parameters, increase digestive activities, and maintain antioxidant capabilities via the Keap1/Nrf2 pathway in black carp. Meanwhile, suitable levels of DCP could retain antioxidant capabilities through the Keap1/Nrf2 pathway and enhance immunity via heightening levels of LZM, ALP, IgM, and *leap-2*. These results supply a corresponding theoretical reference for DCP utilization in the feed of black carp.

## Figures and Tables

**Figure 1 animals-15-01404-f001:**
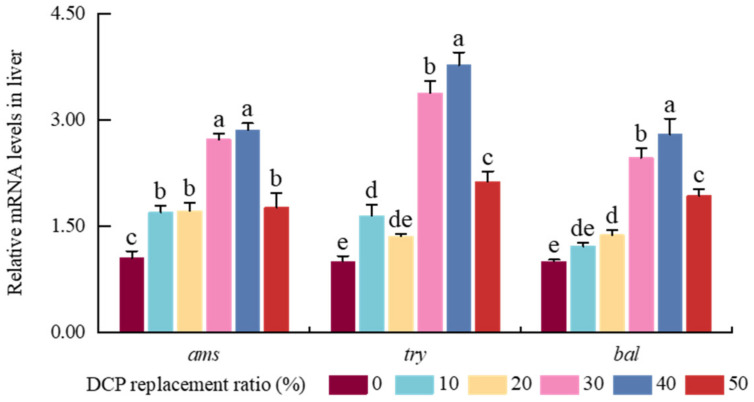
The effects of FM’s replacement with DCP on the expression levels of digestive enzyme-related genes in the liver of black carp. Note: The results were obtained from three independent trials. (Values with different superscripts (a, b, c, etc.) in the same column indicate significant differences (*p* < 0.05), while values sharing at least one common letter (e.g., a and ab) are not significantly different (*p* > 0.05). Different colors indicate different experimental groups. *ams*, amylase; *try*, trypsin; *bal*, bile salt-activated lipase.)

**Figure 2 animals-15-01404-f002:**
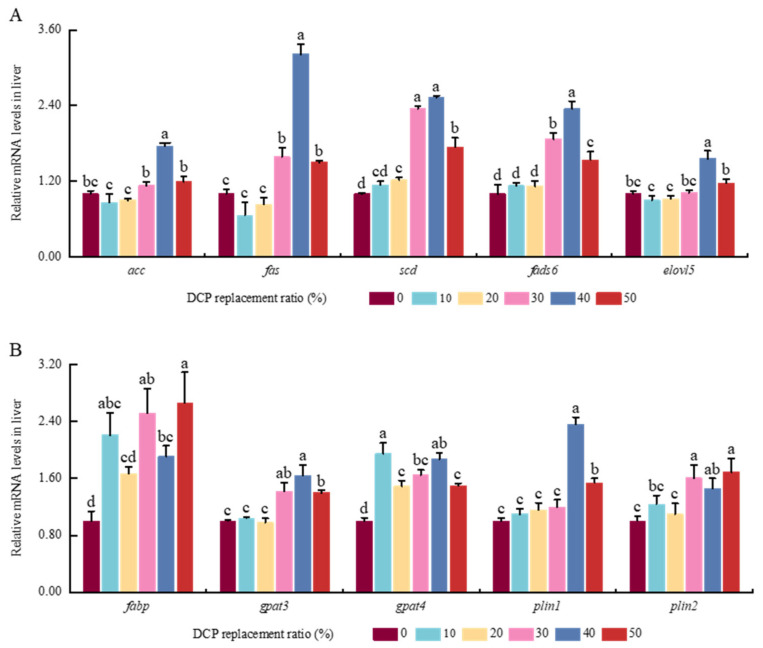
The effects of FM’s replacement with DCP on the expression levels of lipid metabolism-related genes in the liver of black carp. (**A**) Fatty acid synthesis and conversion-related genes; (**B**) fatty acid transport and storage-related genes (Notes: Values with different superscripts (a, b, c, etc.) in the same column indicate significant differences (*p* < 0.05), while values sharing at least one common letter (e.g., a and ab) are not significantly different (*p* > 0.05). Different colors indicate different experimental groups. *acc*, acetyl-CoA carboxylase; *fas*, fatty acid synthase; *scd*, stearoyl-CoA desaturase; *fads6*, fatty acid desaturase 6; *elovl5*, elongation of very long chain fatty acids protein 5; *fabp*, fatty acid-binding protein; *gpat3*, glycerol-3-phosphate acyltransferase 3; *gpat4*, glycerol-3-phosphate acyltransferase 4; *plin1*, perilipin 1; *plin2*, perilipin-2).

**Figure 3 animals-15-01404-f003:**
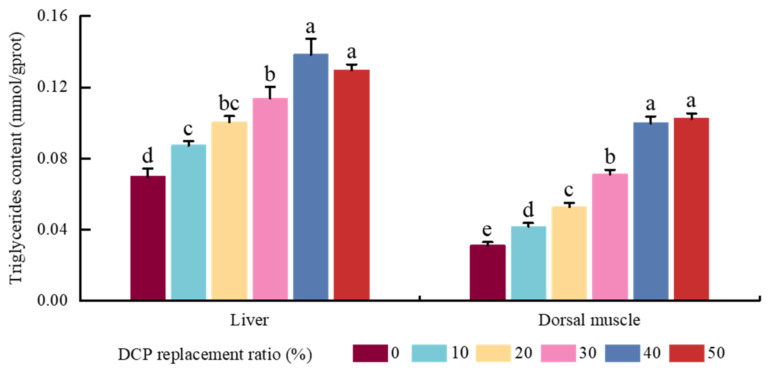
The effects of FM’s replacement with DCP on triglyceride content in the liver and dorsal muscle of black carp. (Notes: Values with different superscripts (a, b, c, etc.) in the same column indicate significant differences (*p* < 0.05), while values sharing at least one common letter (e.g., a and ab) are not significantly different (*p* > 0.05). Different colors indicate different experimental groups.)

**Figure 4 animals-15-01404-f004:**
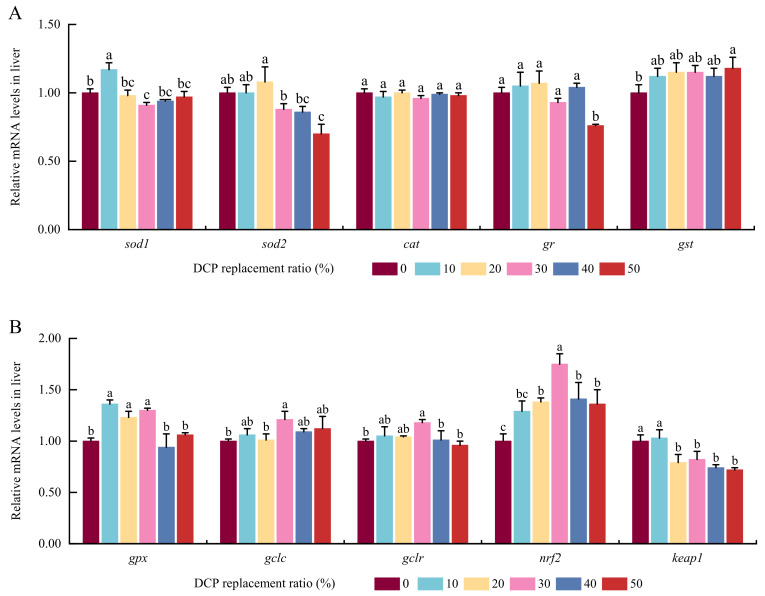
The effects of FM’s replacement with DCP on the expression levels of antioxidant-related genes in the liver of black carp. (**A**) Antioxidant-related genes; (**B**) Glutathione regulation and *nrf2* signal pathway-related genes (Notes: Values with different superscripts (a, b, c, etc.) in the same column indicate significant differences (*p* < 0.05), while values sharing at least one common letter (e.g., a and ab) are not significantly different (*p* > 0.05). Different colors indicate different experimental groups. *sod1*, Cu/Zn superoxide dismutase; *sod2*, Mn superoxide dismutase; *cat*, catalase; *gr*, glutathione reductase; *gst*, glutathione S-transferase; *gpx*, glutathione peroxidase; *gclc*, glutamate–cysteine ligase catalytic subunit; *gclr*, glutamate–cysteine ligase regulatory subunit; *nrf2*, nuclear factor E2-related factor 2; *keap1*, Kelch-like ECH-associated protein 1).

**Figure 5 animals-15-01404-f005:**
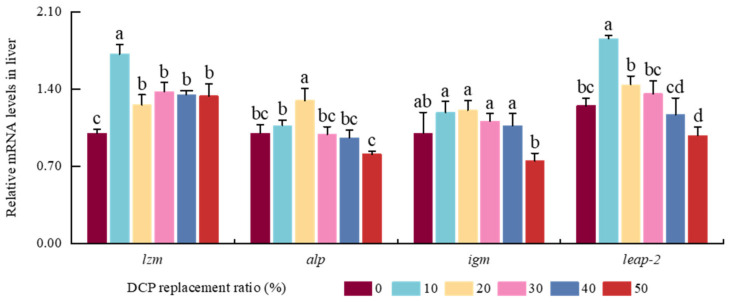
The effects of FM’s replacement with DCP on the expression levels of immunity-related genes in the liver of black carp (Notes: Values with different superscripts (a, b, c, etc.) in the same column indicate significant differences (*p* < 0.05), while values sharing at least one common letter (e.g., a and ab) are not significantly different (*p* > 0.05). Different colors indicate different experimental groups. *lzm*, lysozyme; *alp*, alkaline phosphatase; *igm*, immunoglobulin-binding protein; *leap-2*, liver-expressed antimicrobial peptide 2).

**Table 1 animals-15-01404-t001:** Ingredient and proximate composition of the six experimental diets.

Ingredient (%)	Substitution Ratio of Degossypolled Cottonseed Protein (%)					
	0	10	20	30	40	50
Fishmeal ^A^	40.00	36.00	32.00	28.00	24.00	20.00
DCP ^B^	0.00	4.40	8.80	13.20	17.60	22.00
Soybean meal ^C^	14.00	14.00	14.00	14.00	14.00	14.00
Corn protein meal ^D^	12.00	12.00	12.00	12.00	12.00	12.00
Tapioca flour ^E^	18.00	18.00	18.00	18.00	18.00	18.00
Fish oil ^F^	1.00	1.11	1.23	1.34	1.46	1.57
Rapeseed oil ^G^	1.00	1.11	1.23	1.34	1.46	1.57
Multi-mineral ^H^	2.20	2.20	2.20	2.20	2.20	2.20
Multi-vitamin ^I^	1.10	1.10	1.10	1.10	1.10	1.10
Soy lecithin ^J^	0.40	0.40	0.40	0.40	0.40	0.40
Choline chloride (50%) ^K^	0.40	0.40	0.40	0.40	0.40	0.40
Microcrystalline cellulose ^L^	8.70	8.07	7.44	6.82	6.19	5.56
Total	100.00	100.00	100.00	100.00	100.00	100.00
Nutrient levels (%)						
Crude protein	39.19	37.52	37.32	38.26	37.38	38.97
Crude lipid	6.44	6.41	6.35	6.62	6.48	6.51
Ash	10.44	10.11	9.63	9.16	8.74	8.38

^A^ Guangzhou Muchang import and export trading Co., Ltd., Guangzhou, China. ^B^ Xinjiang Taikun Group Co., Ltd., Xinjiang, China; crude protein: 64%. ^C^ Ningbo Food Co., Ltd., Ningbo, China. ^D^ Henan Julong Bioengineering Co., Ltd., Ruzhou, China. ^E^ Guangxi Honghao Starch Development Co., Ltd., Nanning, China. ^F^ Qingdao Sai Green Marine Biological Feed Co., Ltd., Qingdao, China. ^G^ Yihai Kerry Golden Dragon Fish Food Group Co., Ltd., Shanghai, China. ^H^ KI—2.61 mg, CoCl_2_·6H_2_O—52.44 mg, CuSO_4_·5H_2_O—19.53 mg, FeSO_4_·7H_2_O—347.50 mg, MnSO_4_·H_2_O—45 mg, MgSO_4_·7H_2_O—1279.20 mg, NaCl—127.36 mg, NaSeO_3_—2.63 mg, NaF—2.21 mg, ZnSO_4_·7H_2_O—498.92 mg, Ca(H_2_PO_4_)_2_·H_2_O—15,000.00 mg. ^I^ Vitamin A—20 mg; vitamin D_3_—3 mg; vitamin C—1000 mg; vitamin E—300 mg; thiamine—20 mg; riboflavin—10 mg; pyridoxine hydrochloride—20 mg; vitamin B_12_—0.2 mg; vitamin K_3_—5 mg; inositol—1,000 mg; pantothenic acid—30 mg; folic acid—3 mg; nicotinic acid—50 mg; biotin—1 mg. ^J^ Jiangsu Yuanshengyuan Biological Engineering Co., Ltd., Jiangsu, China; soy lecithin contained 196 mg/g of phosphatidylcholine, 132 mg/g of phosphatidylethanolamine, and 146 mg/g of phosphatidylinositol, with 98% phospholipids. ^K^ Zhejiang Yixing Feed Group Co., Ltd., Jiaxing, China. ^L^ Sinopharm Chemical Reagent Co., Ltd., Shanghai, China.

**Table 2 animals-15-01404-t002:** Amino acid composition of the six experimental diets.

Items (g/100 g)	Substitution Ratio of Degossypolled Cottonseed Protein (%)					
	0	10	20	30	40	50
EAAs						
Thr	1.65	1.54	1.49	1.49	1.53	1.45
Val	2.04	1.97	1.90	1.94	1.97	2.02
Met	1.03	0.80	0.80	0.72	0.82	0.69
Ile	1.76	1.69	1.62	1.64	1.70	1.63
Leu	3.77	3.59	3.46	3.51	3.57	3.48
Phe	1.92	2.03	1.91	2.09	1.93	2.17
Lys	2.45	2.21	2.20	2.16	2.33	1.97
His	1.12	1.09	1.02	1.08	1.09	1.09
Arg	2.36	2.59	2.34	2.75	2.40	3.07
NEAAs						
Asp	3.73	3.60	3.43	3.57	3.55	3.63
Ser	1.75	1.70	1.60	1.66	1.62	1.69
Glu	6.86	7.11	6.53	7.28	6.79	7.84
Gly	2.25	2.09	2.03	2.01	2.12	1.93
Ala	2.74	2.54	2.49	2.44	2.59	2.37
Cys	0.55	0.69	0.59	0.64	0.51	0.78
Tyr	1.46	1.43	1.33	1.38	1.41	1.44
Pro	2.08	2.04	1.91	1.99	2.03	2.02
TEAAs	18.08	17.50	16.72	17.38	17.34	17.58
TAAs	39.49	38.70	36.62	38.34	37.96	39.29

Note: Thr, threonine; Val, valine; Met, methionine; Ile, isoleucine; Leu, leucine; Phe, phenylalanine; Lys, lysine; His, histidine; Arg, arginine; Asp, aspartic acid; Ser, serine; Glu, glutamic acid; Gly, glycine; Ala, alanine; Cys, cysteine; Tyr, tyrosine; Pro, proline; EAAs, essential amino acids; NEAAs, non-essential amino acids; TEAAs, total essential amino acids; TAAs, total amino acids. The following tables are the same.

**Table 3 animals-15-01404-t003:** The list of qPCR primers used in this study.

Gene	Forward (5′–3′)	Reverse (5′–3′)
*ams*	GTTCCCATCAACTCTGACTCTA	GGTTGCTTCCATTGTCCC
*try*	CTCTGAACTCTGGCTACCACTTTT	GCTCAGAACCCTCAGTAACTACGAT
*bal*	GGGAAGCCATTCACCACTC	CCCTTCGGTAAGTGGTGAG
*lzm*	TGCCTTGTTCAGATTTGCT	GTAACTATCCCAGGTGTCCC
*alp*	AGGGTACAGTTGGAGTGAGCG	ACTGATTTGCCTGCGTCTTT
*igm*	CTGGTCTGTACCGCCTCTG	CTGGACTGACTGGGAATAGG
*leap-2*	AAACCTCACGGTGCCTACT	CTCCTGCATATTCCTGTCG
*acc*	GCCAGTCTCCCAACTCCTA	ATGCGATACCTGTCCACCT
*fas*	CATTCCGTAGTAGGATAAGTCAACA	CATAGTCATAACCACGCAGTCG
*scd*	CCCTTCAGCATCTCCTTT	GTGGTAATGTGGCCTTGTA
*fads6*	CATCGGGCTGCCAATGTTT	GTAGCCCGACGGTTACAAA
*elovl5*	AAGAACAGACAGCCATACTCCT	TTCTTGTCTGTCGGTATGAGGA
*fabp*	CTAGGCAAGTAGGCAATGTTA	GATCCGTTCATCCGTTACAAT
*gpat3*	AGAGGGCGATAGTGAGGG	CAGGATGGGAAGTTTGGTC
*gpat4*	CTGTTGTTGGGCTGTTGC	TCCATTCTTGGGTTTATTCTC
*plin1*	CCGTCCCAGTGACAACTCT	GGCAGGGTCACTGTTGAGA
*plin2*	GAGTGGACACGGCCCTAA	GAACCCAGGCGGACATAG
*sod1*	TGAGCAGGAGGGCGATTC	GCACTGATGCACCCATTTGTA
*sod2*	CAGGGATCTACAGGTCTCATT	ACGCTCGCTCACATTCTC
*cat*	ACCTATTGCTGTCCGCTTCTC	TCCCAGTTGCCCTCCTCA
*gr*	ATCACGAGCAGGAAGAGTCAG	CATCTCATCACAGCCCAAGC
*gst*	GAGCCCATCAGAACACCC	ACCCAAATAGCACCCAAC
*gpx*	GCAACCAGTTCGGACATCA	GCGTTCTCACCGTTCACTT
*gclc*	CGGCATCCTTCAGTTCCTC	GCCGTAGGAAGTCAAGGAG
*gclr*	GTCCGTCTTCGCCTAGTGA	CAGGCAGAAGCGGATCACT
*nrf2*	TGTCCAAACACCAGCTCAA	CGTACTCTAGGCCCACGAT
*keap1*	GGCTGCTCTGTGATCTGGTT	TTCCTTGAAGTTGCTGGTGA

*ams*, amylase; *try*, trypsin; *bal*, bile salt-activated lipase; *lzm*, lysozyme; *alp*, alkaline phosphatase; *igm*, immunoglobulin-binding protein; *leap-2*, liver-expressed antimicrobial peptide 2; *acc*, acetyl-CoA carboxylase; *fas*, fatty acid synthase; *scd*, stearoyl-CoA desaturase; *fads6*, fatty acid desaturase 6; *elovl5*, elongation of very long chain fatty acid protein 5; *fabp*, fatty acid-binding protein; *gpat3*, glycerol-3-phosphate acyltransferase 3; *gpat4*, glycerol-3-phosphate acyltransferase 4; *plin1*, perilipin 1; *plin2*, perilipin-2; *sod1*, Cu/Zn superoxide dismutase; *sod2*, Mn superoxide dismutase; *cat*, catalase; *gr*, glutathione reductase; *gst*, glutathione S-transferase; *gpx*, glutathione peroxidase; *gclc*, glutamate–cysteine ligase catalytic subunit; *gclr*, glutamate–cysteine ligase regulatory subunit; *nrf2*, nuclear factor E2-related factor 2; *keap1*, Kelch-like ECH-associated protein 1.

**Table 4 animals-15-01404-t004:** The effects of FM replacement with DCP on the growth performance and feed utilization of black carp.

Items	Substitution Ratio of DCP (%)					
	0	10	20	30	40	50
IBW (g)	7.20 ± 0.10	7.07 ± 0.06	6.93 ± 0.25	7.07 ± 0.21	6.87 ± 0.15	6.77 ± 0.40
FBW (g)	23.03 ± 0.63	21.62 ± 1.08	23.90 ± 1.11	25.81 ± 2.77	25.32 ± 3.01	22.83 ± 2.32
WG (g)	219.89 ± 8.26	205.94 ± 12.83	244.85 ± 17.6	265.86 ± 45.93	268.17 ± 35.81	226.14 ± 33.14
SGR (%)	1.53 ± 0.03	1.47 ± 0.05	1.63 ± 0.07	1.70 ± 0.17	1.71 ± 0.13	1.55 ± 0.14
VSI (%)	5.89 ± 0.13 ^c^	6.09 ± 0.23 ^bc^	6.42 ± 0.38 ^ab^	6.61 ± 0.21 ^a^	6.57 ± 0.19 ^a^	6.84 ± 0.13 ^a^
HSI (%)	1.61 ± 0.05 ^c^	1.67 ± 0.03 ^c^	1.64 ± 0.04 ^c^	1.69 ± 0.02 ^bc^	1.76 ± 0.08 ^ab^	1.80 ± 0.02 ^a^
ISI (%)	0.43 ± 0.06 ^b^	0.46 ± 0.02 ^b^	0.45 ± 0.05 ^b^	0.64 ± 0.07 ^a^	0.61 ± 0.07 ^a^	0.66 ± 0.03 ^a^
CF (g/cm^3^)	1.78 ± 0.05 ^a^	1.65 ± 0.08 ^b^	1.65 ± 0.10 ^b^	1.74 ± 0.05 ^ab^	1.69 ± 0.05 ^ab^	1.62 ± 0.04 ^b^
PER	1.16 ± 0.27	1.21 ± 0.25	1.59 ± 0.41	1.96 ± 0.26	1.92 ± 0.37	1.90 ± 0.29
FI (% BW/day)	5.58 ± 0.05 ^ab^	5.97 ± 0.31 ^a^	5.59 ± 0.43 ^ab^	5.12 ± 0.22 ^bc^	4.82 ± 0.17 ^bc^	4.53 ± 0.35 ^c^
FCR (%)	1.56 ± 0.06	1.71 ± 0.12	1.47 ± 0.09	1.34 ± 0.20	1.36 ± 0.22	1.29 ± 0.20

IBW (g), initial body weight; FBW (g), final body weight; weight gain (WG, %) = (final body weight − initial body weight)/initial body weight × 100; specific growth rate (SGR, %/d) = (ln final body weight − ln initial body weight) × 100/days; viscerosomatic index (VSI, %) = viscera weight/final body weight × 100. Hepatosomatic index (HSI, %) = liver weight/final body weight × 100; intestinalsomatic index (ISI, %) = intestinal fat weight/final body weight × 100; condition factor (CF, %) = final body weight (g)/length (cm)^3^ × 100; protein efficiency ratio (PER) = (final body weight − initial body weight)/protein intake; feed intake (FI) = 100 × total diet intake/[(initial body weight + final body weight)/2 × days]; feed conversion ratio (FCR) = dry feed consumed/(final body weight − initial body weight). Note: The results were obtained from three independent trials. Values in the same row with different lowercase letters (a, b, c, etc.) indicate significant differences (*p* < 0.05), while values sharing at least one common letter (e.g., a and ab) are not significantly different (*p* > 0.05).

**Table 5 animals-15-01404-t005:** The effects of FM replacement with DCP on the body composition of black carp.

Items (%)	Substitution Ratio of DCP (%)					
	0	10	20	30	40	50
Whole fish						
Moisture	76.6 ± 0.34 ^ab^	76.42 ± 0.91 ^ab^	77.06 ± 1.22 ^a^	74.98 ± 0.20 ^ab^	74.43 ± 0.34 ^b^	75.02 ± 1.11 ^ab^
Crude protein	16.25 ± 0.23 ^ab^	16.33 ± 0.12 ^a^	16.04 ± 0.14 ^abc^	15.92 ± 0.17 ^abc^	15.86 ± 0.11 ^bc^	15.75 ± 0.11 ^c^
Crude lipid	3.19 ± 0.24 ^c^	3.38 ± 0.08 ^c^	3.48 ± 0.05 ^c^	5.21 ± 0.35 ^b^	6.48 ± 0.07 ^a^	6.03 ± 0.07 ^a^
Ash	3.37 ± 0.05 ^a^	3.42 ± 0.08 ^a^	3.36 ± 0.06 ^a^	3.22 ± 0.22 ^ab^	2.83 ± 0.03 ^c^	2.95 ± 0.01 ^bc^
Dorsal muscle						
Moisture	79.60 ± 0.59	79.56 ± 0.22	79.79 ± 0.23	79.68 ± 0.27	79.53 ± 0.49	77.83 ± 2.67
Crude protein	19.59 ± 0.04 ^a^	18.07 ± 0.05 ^b^	18.20 ± 0.07 ^c^	18.06 ± 0.03 ^c^	17.99 ± 0.02 ^c^	17.85 ± 0.04 ^d^
Crude lipid	0.67 ± 0.17 ^bc^	0.64 ± 0.07 ^c^	0.69 ± 0.07 ^bc^	0.96 ± 0.04 ^ab^	1.20 ± 0.13 ^a^	1.13 ± 0.12 ^a^
Ash	1.30 ± 0.06	1.26 ± 0.05	1.23 ± 0.04	1.21 ± 0.04	1.20 ± 0.02	1.28 ± 0.02

Note: The results were obtained from three independent trials. Values in the same row with different lowercase letters (a, b, c, etc.) indicate significant differences (*p* < 0.05), while values sharing at least one common letter (e.g., a and ab) are not significantly different (*p* > 0.05).

**Table 6 animals-15-01404-t006:** The effects of FM replacement with DCP on the amino acid composition of the flesh of black carp.

Items (g/100 g)	Substitution Ratio of DCP (%)					
	0	10	20	30	40	50
EAAs						
Thr	3.52 ± 0.17	3.55 ± 0.03	3.23 ± 0.26	3.29 ± 0.22	3.42 ± 0.12	3.28 ± 0.12
Val	4.31 ± 0.17	4.48 ± 0.10	4.14 ± 0.23	4.14 ± 0.26	4.35 ± 0.08	4.22 ± 0.13
Met	2.23 ± 0.13	2.32 ± 0.05	2.13 ± 0.11	2.08 ± 0.12	2.30 ± 0.10	2.06 ± 0.07
Ile	3.92 ± 0.16	4.07 ± 0.09	3.75 ± 0.20	3.74 ± 0.23	3.95 ± 0.06	3.85 ± 0.11
Leu	6.87 ± 0.30	7.08 ± 0.15	6.58 ± 0.46	6.59 ± 0.45	6.88 ± 0.08	6.62 ± 0.22
Phe	3.73 ± 0.18	3.87 ± 0.11	3.59 ± 0.24	3.60 ± 0.25	3.71 ± 0.04	3.58 ± 0.11
Lys	8.49 ± 0.39	8.79 ± 0.22	8.23 ± 0.55	8.26 ± 0.52	8.59 ± 0.11	8.29 ± 0.26
His	2.56 ± 0.10	2.55 ± 0.10	2.47 ± 0.15	2.59 ± 0.14	2.67 ± 0.07	2.63 ± 0.08
Arg	4.68 ± 0.24	4.75 ± 0.10	4.52 ± 0.27	4.54 ± 0.31	4.61 ± 0.10	4.52 ± 0.16
NEAAs						
Asp	8.59 ± 0.39	8.85 ± 0.14	8.23 ± 0.56	8.25 ± 0.52	8.62 ± 0.14	8.29 ± 0.26
Ser	3.10 ± 0.16	3.08 ± 0.08	2.80 ± 0.29	2.90 ± 0.23	2.98 ± 0.12	2.83 ± 0.12
Glu	13.70 ± 0.63	13.95 ± 0.18	12.74 ± 0.89	12.73 ± 0.85	13.45 ± 0.48	13.02 ± 0.55
Gly	3.77 ± 0.19	3.80 ± 0.11	3.59 ± 0.21	3.70 ± 0.39	3.70 ± 0.09	3.49 ± 0.11
Ala	5.27 ± 0.25	5.43 ± 0.13	5.08 ± 0.35	5.15 ± 0.39	5.30 ± 0.07	5.12 ± 0.16
Cys	1.34 ± 0.13	1.42 ± 0.12	1.38 ± 0.11	1.38 ± 0.03	1.14 ± 0.31	1.37 ± 0.06
Tyr	2.92 ± 0.11	3.02 ± 0.05	2.77 ± 0.21	2.79 ± 0.22	2.93 ± 0.05	2.78 ± 0.11
Pro	2.62 ± 0.12	2.64 ± 0.08	2.42 ± 0.18	2.52 ± 0.26	2.64 ± 0.02	2.46 ± 0.11
TEAAs	40.31 ± 1.72	41.47 ± 0.88	38.63 ± 2.44	38.83 ± 2.48	40.50 ± 0.61	39.07 ± 1.21
TAAs	81.61 ± 3.62	83.66 ± 1.24	77.64 ± 5.22	78.26 ± 5.38	81.25 ± 1.61	78.43 ± 2.48

**Table 7 animals-15-01404-t007:** The effects of FM replacement with DCP on the serum biochemical parameters of black carp.

Items	Substitution Ratio of DCP (%)					
	0	10	20	30	40	50
IGF-I (ng/mL)	266.43 ± 15.86 ^ab^	277.49 ± 7.77 ^a^	257.36 ± 12.59 ^abc^	268.91 ± 26.08 ^ab^	220.23 ± 24.34 ^bc^	203.07 ± 28.75 ^c^
IGF-II (ng/mL)	88.86 ± 4.52 ^b^	100.25 ± 8.71 ^ab^	98.6 ± 1.00 ^ab^	104.50 ± 4.26 ^a^	93.99 ± 4.89 ^ab^	90.73 ± 5.01 ^ab^
INS (mIU/L.)	571.11 ± 20.83 ^a^	551.82 ± 15.94 ^a^	553.52 ± 25.46 ^a^	458.61 ± 11.60 ^b^	488.40 ± 7.84 ^b^	442.99 ± 18.72 ^b^
5-HT (ng/mL)	35.93 ± 3.38 ^cd^	33.23 ± 3.06 ^d^	34.1 ± 0.67 ^cd^	40.20 ± 1.50 ^bc^	44.68 ± 2.54 ^b^	52.37 ± 2.22 ^a^
HDL-C (mmol/L)	0.45 ± 0.03	0.45 ± 0.03	0.39 ± 0.04	0.41 ± 0.02	0.42 ± 0.09	0.34 ± 0.03
LDL-C (mmol/L)	1.41 ± 0.10 ^c^	1.55 ± 0.08 ^bc^	1.64 ± 0.11 ^abc^	1.61 ± 0.04 ^bc^	1.87 ± 0.06 ^a^	1.66 ± 0.12 ^ab^
GLU (mmol/L)	3.79 ± 0.52 ^c^	3.45 ± 0.11 ^c^	3.88 ± 0.09 ^bc^	4.78 ± 0.33 ^b^	5.78 ± 0.45 ^a^	5.78 ± 0.30 ^a^
TG (mmol/L)	3.15 ± 0.17 ^b^	2.97 ± 0.07 ^b^	3.13 ± 0.04 ^b^	3.28 ± 0.22 ^b^	4.20 ± 0.21 ^a^	3.91 ± 0.11 ^a^
TC (mmol/L)	3.94 ± 0.24 ^bc^	4.01 ± 0.16 ^ab^	3.71 ± 0.17 ^bc^	3.72 ± 0.08 ^bc^	4.40 ± 0.06 ^a^	3.53 ± 0.18 ^c^
AST (U/L)	183.27 ± 3.19 ^ab^	198.67 ± 7.95 ^a^	159.20 ± 14.88 ^bc^	135.83 ± 15.61 ^c^	153.33 ± 6.63 ^c^	163.57 ± 8.83 ^bc^
ALT (U/L)	6.57 ± 0.58 ^ab^	7.93 ± 1.58 ^a^	5.10 ± 1.31 ^b^	4.13 ± 0.95 ^b^	4.40 ± 0.17 ^b^	6.20 ± 0.70 ^ab^
ALP (U/L)	69.17 ± 5.58 ^b^	76.13 ± 2.12 ^ab^	77.8 ± 5.37 ^ab^	77.7 ± 3.00 ^ab^	85.90 ± 9.70 ^a^	78.07 ± 4.27 ^ab^
ALB (g/L)	10.27 ± 0.50 ^a^	10.17 ± 0.40 ^ab^	10.13 ± 0.31 ^ab^	9.27 ± 0.23 ^bc^	10.00 ± 0.30 ^ab^	9.20 ± 0.10 ^c^
TBA (umol/L)	5.97 ± 0.15 ^b^	5.98 ± 0.71 ^b^	6.20 ± 0.46 ^ab^	6.13 ± 0.31 ^ab^	7.00 ± 0.26 ^ab^	7.40 ± 0.82 ^a^
BUN (mmol/L)	1.84 ± 0.09 ^c^	1.93 ± 0.23 ^c^	2.15 ± 0.11 ^bc^	2.24 ± 0.10 ^bc^	2.75 ± 0.16 ^a^	2.53 ± 0.14 ^ab^
LDH (U/L)	1122.37 ± 87.11 ^a^	1041.6 ± 127.81 ^a^	786.17 ± 120.28 ^b^	753.1 ± 65.73 ^b^	916.57 ± 18.47 ^ab^	1037.87 ± 82.47 ^a^

Note: The results were obtained from three independent trials. Values in the same row with different lowercase letters (a, b, c, etc.) indicate significant differences (*p* < 0.05), while values sharing at least one common letter (e.g., a and ab) are not significantly different (*p* > 0.05).

**Table 8 animals-15-01404-t008:** The effects of FM replacement with DCP on digestive enzymes in the liver of black carp.

Items (U/mg prot)	Substitution Ratio of DCP (%)					
	0	10	20	30	40	50
α-AMS	0.55 ± 0.05 ^e^	0.74 ± 0.02 ^d^	1.08 ± 0.06 ^c^	1.38 ± 0.05 ^b^	1.58 ± 0.08 ^a^	1.32 ± 0.14 ^b^
TRY	101.06 ± 5.95 ^d^	126.66 ± 7.96 ^d^	248.89 ± 21.67 ^c^	387.10 ± 56.2 ^b^	479.20 ± 14.10 ^a^	291.85 ± 43.96 ^c^
LPS	1.83 ± 0.16 ^b^	1.90 ± 0.17 ^ab^	2.18 ± 0.20 ^ab^	2.44 ± 0.14 ^ab^	2.54 ± 0.39 ^ab^	2.86 ± 0.75 ^a^

Note: The results were obtained from three independent trials. Values in the same row with different lowercase letters (a, b, c, etc.) indicate significant differences (*p* < 0.05), while values sharing at least one common letter (e.g., a and ab) are not significantly different (*p* > 0.05).

**Table 9 animals-15-01404-t009:** The effects of FM’s replacement with DCP on antioxidative indices in the liver of black carp.

Items	Substitution Ratio of DCP (%)					
	0	10	20	30	40	50
T-SOD (U/mg prot)	25.86 ± 1.21 ^d^	31.46 ± 0.38 ^cd^	31.84 ± 0.63 ^c^	37.66 ± 2.74 ^ab^	41.40 ± 3.79 ^a^	35.22 ± 1.67 ^bc^
CAT (U/mg prot)	1.31 ± 0.10 ^b^	1.45 ± 0.28 ^ab^	1.71 ± 0.12 ^ab^	1.72 ± 0.06 ^ab^	1.74 ± 0.28 ^ab^	2.01 ± 0.29 ^a^
GPx (U/mg prot)	22.11 ± 1.63 ^a^	21.15 ± 0.34 ^a^	20.65 ± 0.46 ^a^	18.26 ± 0.67 ^b^	15.5 ± 0.33 ^c^	15.18 ± 0.25 ^c^
GSH (mmol/g prot)	586.26 ± 80.23 ^b^	606.35 ± 51.07 ^ab^	675.87 ± 48.68 ^ab^	750.93 ± 54.06 ^a^	669.37 ± 63.18 ^ab^	596.39 ± 42.11 ^ab^
GR (U/mg prot)	0.17 ± 0.03	0.17 ± 0.01	0.22 ± 0.07	0.23 ± 0.10	0.11 ± 0.02	0.12 ± 0.04
GST (U/mg prot)	5.84 ± 0.22 ^b^	5.86 ± 0.42 ^b^	5.92 ± 0.43 ^b^	5.54 ± 0.79 ^b^	6.28 ± 0.52 ^ab^	7.64 ± 0.96 ^a^
T-AOC (mmol/g prot)	1.20 ± 0.10 ^a^	1.11 ± 0.06 ^ab^	0.94 ± 0.03 ^c^	0.85 ± 0.07 ^c^	0.98 ± 0.03 ^bc^	0.82 ± 0.03 ^c^
H_2_O_2_ (mmol/g prot)	24.92 ± 0.19 ^b^	21.03 ± 0.41 ^d^	22.20 ± 0.17 ^c^	17.63 ± 0.27 ^e^	22.62 ± 0.27 ^c^	27.60 ± 0.20 ^a^
MDA (nmol/g prot)	0.66 ± 0.1 ^c^	0.71 ± 0.06 ^bc^	0.53 ± 0.07 ^c^	0.91 ± 0.06 ^ab^	1.06 ± 0.11 ^a^	0.96 ± 0.08 ^a^

Note: The results were obtained from three independent trials. Values in the same row with different lowercase letters (a, b, c, etc.) indicate significant differences (*p* < 0.05), while values sharing at least one common letter (e.g., a and ab) are not significantly different (*p* > 0.05).

**Table 10 animals-15-01404-t010:** The effects of FM’s replacement with DCP on immunity in the liver and intestine of black carp.

Items	Substitution Ratio of DCP (%)					
	0	10	20	30	40	50
Liver						
LZM (U/mg prot)	50.89 ± 0.21 ^c^	54.28 ± 0.38 ^a^	52.80 ± 0.37 ^b^	54.57 ± 0.40 ^a^	53.29 ± 0.12 ^b^	48.94 ± 0.34 ^d^
ALP (U/g prot)	0.21 ± 0.01 ^cd^	0.23 ± 0.02 ^bc^	0.32 ± 0.01 ^a^	0.26 ± 0.04 ^b^	0.26 ± 0.01 ^b^	0.19 ± 0.01 ^d^
IgM (mg/mg prot)	2.52 ± 0.10 ^b^	2.86 ± 0.20 ^a^	2.95 ± 0.09 ^a^	2.29 ± 0.17 ^b^	2.3 ± 0.04 ^b^	2.29 ± 0.08 ^b^
Intestine						
LZM (U/mg prot)	18.69 ± 0.15 ^c^	20.31 ± 0.1 ^b^	20.56 ± 0.19 ^b^	22.49 ± 0.21 ^a^	22.53 ± 0.18 ^a^	23.49 ± 1.11 ^a^
ALP (U/g prot)	12.74 ± 0.27 ^c^	12.93 ± 0.67 ^c^	16.04 ± 0.50 ^a^	16.16 ± 0.69 ^a^	14.83 ± 1.17 ^ab^	13.44 ± 0.37 ^bc^
IgM (mg/mg prot)	2.49 ± 0.06 ^a^	2.48 ± 0.17 ^a^	2.36 ± 0.09 ^a^	2.40 ± 0.09 ^a^	2.33 ± 0.05 ^a^	1.92 ± 0.09 ^b^

Note: The results were obtained from three independent trials. Values in the same row with different lowercase letters (a, b, c, etc.) indicate significant differences (*p* < 0.05), while values sharing at least one common letter (e.g., a and ab) are not significantly different (*p* > 0.05).

## Data Availability

The data presented in this study are available in the main article.

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
