# Peer review of "The Effects of Fishmeal Replacement with Degossypolled Cottonseed Protein on Growth, Serum Biochemistry, Endocrine Responses, Lipid Metabolism, and Antioxidant and Immune Responses in Black Carp (*Mylopharyngodon piceus*)"

_animals, 2025, doi:10.3390/ani15101404_

Round 1

Reviewer 1 Report

Comments and Suggestions for Authors

The manuscript is generally well-composed, and the methods utilized are robust. Nevertheless, it is not ready for publication in its present state due to several minor issues outlined below.

  1. The Abstract should include: "DCP replacing fishmeal with substitution amounts of 0%, 10%, 20%, 30%, 40%, and 50% (DCP0, DCP10, DCP20, DCP30, DCP40, DCP50), respectively."
  2. The keyword "muscle quality" is not highly relevant to the full text.
  3. Lines 80-81 should explicitly mention the production of black carp in China.
  4. Clarify in lines 99-101 whether the feed is sinking or floating.
  5. In Table 1, state the protein content of fish meal, and explain the difference of more than 1.5% in protein levels across the six feed types.
  6. Clarify the logic of sample handling: weigh first, then sample.
  7. Correct line 181 to "Measurement of relative gene expression."
  8. In lines 203-206, specify whether "standard deviation or standard error?" was used, and note that the homogeneity of variances was assessed before data analysis.
  9. Ensure gene names (relative gene expression) are italicized in the manuscript.
  10. The discussion section misses key results, such as: Line 361-362, is gossypol protein dephenolized? Line 388-390, supplemental carbohydrate content; Line 456-458, missing liver section results.
  11. Remove the comma in line 366.
  12. Reference citations should be accurate and their number should be appropriately reduced.
Comments on the Quality of English Language

Can be appropriately raised

Author Response

Question 1: The manuscript is generally well-composed, and the methods utilized are robust. Nevertheless, it is not ready for publication in its present state due to several minor issue outline below.

Answer: We sincerely thanked the reviewer for the positive evaluation of our manuscript. We also appreciated your constructive suggestions and comments. We fully agree that addressing them is essential for improving the clarity and quality of the manuscript. Accordingly, we have carefully revised the manuscript to address each point as outlined below.

Question 2: The Abstract should include:"DCP replacing fishmeal with substitution amounts of 0%, 10%, 20%, 30%, 40%, 50% (DCP0, DCP10, DCP20, DCP30, DCP40 and DCP50), respectively."

Answer: We sincerely thanked the reviewer’s suggestion and comments. We have added relevant content in the new manuscript (Lines 25-27).

Question 3: The keyword "muscle quality" is not highly relevant to the full text.

Answer: We sincerely thanked the reviewer’s suggestion and comments. We sincerely apologize for the error. We have deleted "muscle quality" in the new manuscript (Line 45).

Question 4: Lines 80-81 should explicitly mention the production of black carp in China.

Answer: We sincerely thanked the reviewer’s suggestion and comments. We have received Line 84 to include the specific production data of black carp in China.

Question 5: Clarify in lines 99-101 whether the feed is sinking or floating.

Answer: We sincerely thanked the reviewer’s suggestion and comments. We have clarified in Line 107 that the experimental feed used in this study was of the sinking type.

Question 6: In Table 1, state the protein content of fish meal, and explain the difference of more than 1.5% in protein levels across the six feed types.

Answer: We sincerely thanked the reviewer’s suggestion and comments. The protein contents presented in Table 1 are based on actual measurements of the experimental diets. We acknowledge that the discrepancies may have arisen due to errors during the manual weighing of raw materials or insufficient uniformity during mechanical mixing in the feed preparation process. Although we followed strict protocols, such variations are still possible in small-scale experimental settings. We sincerely apologize for the error and appreciate the reviewer’s understanding.

Question 7: Clarify the logic of sample handling: weigh first, then sample.

Answer: We sincerely thanked the reviewer’s suggestion and comments. We have clarified the sample handling procedure in the revised manuscript to indicate that the fish were first weighed and then sampled accordingly (Lines 166-169).

Question 8: Correct line 181 to "Measurement of relative gene expression."

Answer: We sincerely thanked the reviewer’s suggestion and comments. We have modified the title of this section to "Measurement of relative gene expression." (Line 203).

Question 9: In lines 203-206, specify whether "standard deviation or standard error?" was used, and note that the homogeneity of variances was assessed before data analysis.

Answer: We sincerely thanked the reviewer’s suggestion and comments. In the received manuscript, we made it clear that the data are expressed as mean ± standard deviation (Mean ± SD), and added a statement noting that homogeneity of variances was assessed using Levene’s test prior to ANOVA analysis (Lines 224-229).

Question 10: Ensure gene names (relative gene expression) are italicized in the manuscript.

Answer: We sincerely thanked the reviewer’s suggestion and comments. In the received manuscript, we carefully checked the manuscript and changed the name of all relative gene names to italic.

Question 11: The discussion section misses key results, such as: Line 361-362, is gossypol protein dephenolized? Line 388-390, supplemental carbohydrate comtent; Line 456-458, missing liver section results.

Answer: We sincerely thanked the reviewer’s suggestion and comments. To improve the rigor of the manuscript, we have clarified that was low-gossypol cottonseed meal in Line 473. We have added the carbohydrate content (19.30 ± 0.1%) in Line 544. Although liver histological sections were collected and examined, the changes observed were not sufficiently representative or distinct to warrant the eventual choice to remove this section from this manuscript. Moreover, the key findings of the study are already well supported by biochemical and gene expression data. Therefore, we have removed the reference to liver sections in Lines 623–624 to maintain focus and clarity.

Question 12: Remove the comma in line 366.

Answer: We sincerely thanked the reviewer’s suggestion and comments. We have removed the comma in Line 479.

Question 13: Reference citations should be accurate and their number should be appropriately reduced.

Answer: We sincerely thanked the reviewer’s suggestion and comments. We have carefully reviewed and revised the reference citations throughout the manuscript. Inappropriate or redundant citations have been removed, and the remaining references have been checked for accuracy and relevance to ensure that each citation directly supports the associated content.

Reviewer 2 Report

Comments and Suggestions for Authors

The manuscript provides a basis for future studies on alternative protein sources, particularly on using degossypolled cottonseed protein as a substitute for fishmeal in aquaculture. The Introduction gives enough background for the research. The M&M section is sufficient. Please check carefully results description to avoid mistakes. I noticed that MDA results in L312-313 do not match the Table. Overall, the article makes good impression but it is necessary to check grammar throughout the text, Results and Discussion especially. Some minor issues which are listed below should be addressed.

L28 please replace “these six” with another word like test groups or else.

L28 and throughout the Abstract, please provide full names of enzymes when you use the abbreviation for the first time.

L31 please correct “ALTD, CP40”

L32 “lower-levels”, remove hyphen

L33 increased

L47, 84, 90, etc. Use FM instead of fishmeal since you introduce the abbreviation on L44.

L73 seabass

L93-94 please rephrase to enhance clarity

L140 Section 2.3 – check grammar please

L203 check grammar please

L216-217 please rephrase to enhance clarity

L218 shouldn’t it be “increase” instead of “improvement”?

L246 please check the spelling

L246 and Tables 6-7 Is it necessary to divide hormones and “biochemistry”? In my opinion, these serum parameters can be combined in one table.

L272 activity

L308 please check the spelling

L312 DCP0 did not have the highest MDA concentration. I’d say that it was elevated in DCP30-50 compared with DCP0-20.

L354 Did you mean statistically significant differences? Please rephrase to enhance clarity.

L356-357 No logical connection between these sentences. Extra efforts to make the text more fluent would be beneficial.

L360 change FM group to DCP0 group (to match the previous text)

L362 please check the spelling

L366 redundant commas

L400 and DCH40 group

L456 please check the spelling

L465 “These antioxidant enzymes (SOD, CAT and GPx) can catalyze O2‾ and H2O2, respectively“. This is not correct, please rephrase, because superoxide dismutase catalyzes the dismutation of the superoxide anion radical into molecular oxygen and hydrogen peroxide, catalase catalyzes the decomposition of hydrogen peroxide to water and oxygen, and glutathione peroxidase reduces lipid hydroperoxides to their corresponding alcohols and free hydrogen peroxide to water.

L475 confusing sentence, please rephrase. Elevated antioxidant enzyme activities does not alleviate ROS damage, they protect cells from it.

L480, L498 What does “adequate DCP groups” mean here exactly?

L504-505 in hybrid grouper [11] and hybrid grouper [101] – please make correction

L514 – please delete

L520 Isn’t biochemical “parameters” a more suitable term for “biochemical indices, serum hormones’ levels”?

L524 DCP

Comments on the Quality of English Language

It is necessary to check grammar throughout the text, Results and Discussion especially.

Author Response

Question 1: The manuscript provides a basis for future studies on alternative protein sources, particularly on using degossypolled cottonseed protein as a substitute for fishmeal in aquaculture. The introduction gives enough background for the research. The M&M section is sufficient. Please check carefully results description to avoid mistakes. I noticed that MDA results in L312-313 do not match the Table. Overall, the article makes good impression but it is necessary to check grammar throughout the text, Results and Discussion especially. Some minor issues which are listed below should be addressed.

Answer: We sincerely thanked you for the positive and encouraging comments. We appreciated your recognition of the significance of our study and the soundness of the introduction and M&M sections. We have carefully checked the entire manuscript, particularly the results description, and revised the text to ensure grammatical correctness and consistency between the described results and the tables. We have corrected the inconsistency in the MDA results mentioned in Lines 392–393, so that they now match the Table. Additionally, all issues and suggestions raised by you have been addressed point by point below. We are grateful for your thoughtful review, which helped us improve the clarity of our manuscript.

Question 2: L28 please replace “these six” with another word like test groups or else.

Answer: We sincerely thanked the reviewer’s suggestion and comments. We have replaced “these six” with test in Line 28.

Question 3: L28 and throughout the Abstract, please provide full names of enzymes when you use the abbreviation for the first time.

Answer: We sincerely thanked the reviewer’s suggestion and comments. In the received manuscript, we have provided full names of enzymes when we use the abbreviation for the first time. 

Question 4: L31 please correct “ALTD, CP40”

Answer: We sincerely thanked the reviewer’s suggestion and comments. Thank you for pointing this out. We have corrected the terms in this section to “ALT, DCP40” to ensure accuracy and consistency.

Question 5: L32“lower-levels", remove hyphen

Answer: We sincerely thanked the reviewer’s suggestion and comments. We have removed the hyphen in Line 32.

&nbsp

Question 6: L33 increased

Answer: We sincerely thanked the reviewer’s suggestion and comments. We have corrected the grammatical issue in Line 32.

Question 7: L47, 84, 90, etc. Use FM instead of fishmeal since you introduce the abbreviation on L44.

Answer: We sincerely thanked the reviewer’s suggestion and comments. We have replaced “fishmeal” with “FM” in Lines 51, 87, 99, and throughout the rest of the text to maintain consistency with the abbreviation introduced in Line 48.

Question 8: L73 seabass

Answer: We sincerely thanked the reviewer’s suggestion and comments. We were sorry for our careless mistakes. The spelling error in Line 76 has been corrected in the revised manuscript.

Question 9: L93-94 please rephrase to enhance clarity

Answer: We sincerely thanked the reviewer’s suggestion and comments. We have rephrased Lines 103–104 to improve clarity and completeness.

Question 10: L140 Section 2.3 – check grammar please

Answer: We sincerely thanked the reviewer’s suggestion and comments. We have corrected the grammatical errors in section 2.3 (Line 165).

Question 12: L203 check grammar please.

Answer: We sincerely thanked the reviewer’s suggestion and comments. We have corrected the grammatical errors in Line 224.

Question 13: L216-217 please rephrase to enhance clarity

Answer: We sincerely thanked the reviewer’s suggestion and comments. We have rephrased Lines 240–241 to improve clarity and ensure grammatical accuracy.

Question 14: L218 shouldn't it be “increase” instead of “improvement”?

Answer: We sincerely thanked the reviewer’s suggestion and comments. We have replaced “improvement” with “increase” in Line 241.

Question 15: L246 please check the spelling

Answer: We sincerely thanked the reviewer’s suggestion and comments. We have corrected the spelling errors in Line 273.

Question 16: L246 and Tables 6-7 Is it necessary to divide hormones and “biochemistry?” In my opinion, these serum parameters can be combined in one table.

Answer: We sincerely thanked the reviewer’s suggestion and comments. We have combined the serum hormones and serum biochemistry into a single table (now Table 7).

Question 17: L272 activity

Answer: We sincerely thanked the reviewer’s suggestion and comments. We have corrected the spelling errors in Line 305.

Question 18: L308 please check the spelling

Answer: We sincerely thanked the reviewer’s suggestion and comments. The typographical error “shiwere” has been corrected to “were” in Line 387.

Question 19: L312 DCP0 did not have the highest MDA concentration. I’d say that it was elevated in DCP30-50 compared with DCP0-20

Answer: We sincerely thanked the reviewer’s suggestion and comments. We have revised the description in Lines 392-393 to accurately reflect the MDA concentration trend. The revised manuscript indicated that MDA was lowest in the DCP20 group and significantly increased in the DCP40 and DCP50 groups, consistent with the reported data.

Question 20: L354 Did you mean statistically significant differences? Please rephrase to enhance clarity.

Answer: We sincerely thanked the reviewer’s suggestion and comments. We have rephrased the sentence in Line 463 to clarify that the observed differences were not statistically significant.

Question 21: L356-357 No logical connection between these sentences. Extra efforts to make the text more fluent would be beneficial.

Answer: We sincerely thanked the reviewer’s suggestion and comments. To improve the logical flow and readability of the text, we have revised this part by adding a transition sentence that links growth performance with digestive enzyme activity in Lines 464-468.

Question 22: L360 change FM group to DCP0 group (to match the previous text)

Answer: We sincerely thanked the reviewer’s suggestion and comments. We have changed “FM group” to “DCP0 group” in Line 469 to match the previous text.

Question 23: L362 please check the spelling

Answer: We sincerely thanked the reviewer’s suggestion and comments. We have corrected the spelling error in Line 417.

Question 24: L366 redundant commas

Answer: We sincerely thanked the reviewer’s suggestion and comments. We have deleted the redundant comma in Line 474.

Question 25: L400 and DCH40 group

Answer: We sincerely thanked the reviewer’s suggestion and comments. We have revised the sentence in Line 556 to include the DCP40 group.

Question 26: L456 please check the spelling

Answer: We sincerely thanked the reviewer’s suggestion and comments. We have corrected the spelling error in Line 623.

Question 27: L465“These antioxidant enzymes (SOD, CAT and GPx) can catalyze O2 and H2O2, respectively“. This is not correct, please rephrase, because superoxide dismutase catalyzes the dismutation of the superoxide anion radical into molecular oxygen and hydrogen peroxide, catalase catalyzes the decomposition of hydrogen peroxide to water and oxygen, and glutathione peroxidase reduces lipid hydroperoxides to their corresponding alcohols and free hydrogen peroxide to water.

Answer: We sincerely thanked the reviewer’s suggestion and comments. We have revised Lines 633-636 to provide a scientifically accurate description of the catalytic roles of SOD, CAT, and GPx.

Question 28: L475 confusing sentence, please rephrase. Elevated antioxidant enzyme activities does not alleviate ROS damage, they protect cells from it.

Answer: We sincerely thanked the reviewer’s suggestion and comments. We have revised the sentence in Lines 668-669 to improve clarity and scientific accuracy.

Question 29: L480, L498 What does “adequate DCP groups” mean here exactly?

Answer: We sincerely thanked the reviewer’s suggestion and comments. We have clarified the substitution ratio of DCP in Line 674 and Line 691 to make the expression of the manuscript more scientific and accurate.

Question 30: L504-505 in hybrid grouper [11] and hybrid [101] – please make correction

Answer: We sincerely thanked the reviewer’s suggestion and comments. The two mentioned groups are different hybrid strains (Epinephelus fuscoguttatus ♀ × Epinephelus lanceolatus♂, and Epinephelus fuscoguttatus♀×Epinephelus lanceolatus♂). In the revised manuscript, we have removed the original reference to “hybrid [101]” to streamline the reference list and avoid redundancy. We appreciate your careful reading and constructive feedback, which helped us improve the clarity and consistency of the manuscript.

Question 31: L514 - please delete

Answer: We sincerely thanked the reviewer’s suggestion and comments. We sincerely and deeply apologize for this careless error. The paragraph in question was unintentionally retained from the downloadable article template provided by the journal. During the revision process, we neglected to remove this section. We have now carefully deleted it from the revised manuscript. Thank you again for your careful review and valuable comments.

Question 32: L520 Isn’t biochemical “parameters” a more suitable term for “biochemical indices, serum hormones’ levels”?

Answer: We sincerely thanked the reviewer’s suggestion and comments. We have replaced “biochemical indices and serum hormone levels” with “biochemical parameters” in Line 708 to improve conciseness and scientific accuracy.

Question 33: L524 DCP

Answer: We sincerely thanked the reviewer’s suggestion and comments. We have modified the error in Line 712.

Reviewer 3 Report

Comments and Suggestions for Authors

Dear Authors,

[General Comments]
The manuscript investigates the effects of replacing fishmeal with degossypolled cottonseed protein (DCP) on the growth, serum biochemistry, hormone indices, lipid metabolism, antioxidant, and immune responses in black carp (Mylopharyngodon piceus). The study provides valuable insights into alternative protein sources for aquafeeds. However, the manuscript has several weaknesses that need to be addressed before acceptance. Key concerns include unclear methodological details, inconsistent data interpretations, and lack of discussion on the potential anti-nutritional effects of DCP. Additionally, grammatical errors and improper sentence structures hinder readability.

[Major Comments]
Comment1. Methodology clarity (lines 91–137)
- The experimental design lacks details on how fish were allocated to different treatment groups. How was randomization ensured? Was there any control for potential tank effects?
- The composition of experimental diets is well-documented (Tables 1 and 2); however, the preparation method of the diets is unclear. How was the DCP processed before inclusion in the diets?

Comment 2. Data interpretation issues (lines 208–240, 247–270, and 303–350)
- The results indicate that DCP replacement had no significant effects on weight gain (WG) or specific growth rate (SGR). However, the discussion suggests that certain replacement levels (DCP30, DCP40) improved growth performance. These statements appear contradictory and should be clarified.
- Serum IGF-1 levels decreased with higher DCP inclusion (Lines 247–250). The authors state that DCP improves growth performance, but lower IGF-1 levels suggest otherwise. This inconsistency should be addressed.
- The manuscript claims that lipid metabolism was affected by DCP inclusion, but there is no direct measurement of hepatic lipid deposition or fatty acid composition. Including histological liver analysis or lipid profiling would strengthen the argument.

Comment 3. Potential anti-nutritional factors (lines 60–78, and 374–403)
- The introduction briefly mentions anti-nutritional factors (ANFs) such as free gossypol and tannins. However, their potential residual effects on fish health and metabolism are not discussed in the results. Have the authors confirmed the absence of ANFs in the diets?
- Liver function markers (AST, ALT) fluctuated across treatments (Lines 261–270). Could this be linked to residual gossypol toxicity? A more in-depth discussion is necessary.

Comment 4. Statistical concerns (Lines 202–206, and Tables 3–10)
- Some reported p-values and significance levels appear inconsistent (e.g., Table 3: VSI shows significant differences, but no clear trend is described). Were multiple comparison tests performed?
- Tukey’s post-hoc test is mentioned, but the justification for its use over other methods (e.g., Bonferroni correction) is missing. A brief explanation of statistical choices would enhance clarity.

Comment 5. Language and readability (entire manuscript)
- There are frequent grammatical errors and awkward sentence constructions throughout the manuscript. For example, in the abstract (Lines 25–40), "Although DCP30 could lower activities of AST and ALTD, CP40 could heighten levels of LDL-C, TG, TC, ALP and BUN." The phrase is unclear and should be restructured for better readability.
- The manuscript would benefit from thorough language editing to improve readability and coherence.

[Minor Comments]
Title and keywords
- The title is clear, but "hormone indices" could be more specific. Consider rewording to "endocrine responses" for clarity.
- Keywords should include "anti-nutritional factors" and "feed efficiency" to improve searchability.

Figures and Tables
- Figure 1 (Digestive enzyme gene expression) and Figure 2 (Lipid metabolism gene expression) lack clear legends. Define abbreviations used in figure labels.
- Table 5 (Amino acid composition) should indicate whether differences are statistically significant.

References
- Several citations in the introduction (e.g., Lines 44–49) refer to older studies. Consider incorporating more recent references (post-2020) on plant protein alternatives in aquafeeds.

Comments on the Quality of English Language

There are frequent grammatical errors and awkward sentence constructions throughout the manuscript. The manuscript would benefit from thorough language editing to improve readability and coherence.

Author Response

Question 1: The manuscript investigates the effects of replacing fishmeal with deossyolld cottonseed protein (DCP) on the growth, serum biochemistry, hormone indices, lipid metabolism, antioxidant, and immune responses in black carp (Mylopharyngodon piceus). The study provides valuable insights into alternative protein sources for aquafeeds. However, the manuscript has several weaknesses that need to be addressed before acceptance. Key concerns include unclear methodological details. inconsistent data interpretations, and lack of discussion on the potential anti-nutritional effects of DCP. Additionally, grammatical errors and improper sentence structures hinder readability.

Answer: We sincerely thank the reviewer for the thorough evaluation and constructive comments on our manuscript. We truly appreciate your recognition of the value and relevance of our study regarding the use of deossypolled cottonseed protein (DCP) as an alternative to fishmeal in black carp diets. We acknowledge the concerns raised about methodological clarity, data interpretation, and the insufficient discussion on the potential anti-nutritional effects of DCP. In response, we have carefully revised the manuscript to clarify the experimental procedures, improve the interpretation of results, and include a more comprehensive discussion on possible anti-nutritional factors associated with DCP (Lines 474-476, 568-571). Furthermore, we have thoroughly edited the manuscript to correct grammatical errors and improve sentence structure for better readability and coherence. We are grateful for your insightful feedback, which has significantly improved the overall quality and clarity of the manuscript.

Question 2: Comment1. Methodology clarity (lines 91-137)

-The experimental design lacks details on how fish were allocated to different treatmment groups. How was randomization ensured? Was there any control for potential tank effects?

-The composition of experimental diets is well-documented (Tables 1 and 2); however, the preparation method of the diets is unclear. How was the DCP processed before inclusion in the diets?

Answer: We sincerely thanked the reviewer’s suggestion and comments. 

-In the revised manuscript, we have added a detailed explanation of the experimental design, including the fish allocation method. In our study, randomization was ensured by randomly netting fish from the holding tank, and then randomly selecting individuals of uniform size and good health. These selected fish were then randomly allocated into different experimental groups to ensure similar initial body weight across treatments and to minimize selection bias. (Lines 149-153, Lines 154-156).

-We have clarified the diet preparation process in the revised manuscript (Lines 107-111).

Question 3: Data interpretation issues (linea 208-240, 247-270, and 303-350)

-The results indicate that DCP replacement had no significant effects on weight gain (WG) or specific growth rate (SGR). However, the discussion suggests that certain replacement levels (DCP30, DCP40) improved growth performance. These statements appear contradictory and should be clarified.

-Serum IGF-1 lvels derased with higher DCP inclusion (Lines 247- 250). The authors state that DCP improves growth performance, but lower IGF-1 levels suggest otherwise. This inconsistency should be addressed.

-The manuscript claims that lipid metabolism was affected by DCP inclusion, but there is no direct measurement of hepatic lipid deposition or fatty acid composition. Including histological liver analysis or lipid profiling would strengthen the argument.

Answer: We sincerely thanked the reviewer’s suggestion and comments. 

-In the Results section, we stated that DCP replacement had no statistically significant effects on WG or SGR among the treatment groups. However, we observed that both WG and SGR values showed an increasing trend in the DCP30 and DCP40 groups compared to the DCP0 group. Based on this numerical improvement, we used the term “improved growth performance” in the discussion to highlight this trend, while still acknowledging that the difference was not statistically significant. To avoid confusion, we have revised the relevant sentences in the discussion to clearly indicate that the observed improvement refers to a non-significant trend rather than a significant effect.

-Sincere thanks again for your insightful comment. We agree that serum IGF-1 levels decreased in the DCP40-50 groups, as noted in Table 6. But we believe that growth in animals is influenced by multiple factors and regulated by a range of hormones. IGF-1 is only one of these regulatory elements and may not fully reflect or determine the overall growth performance.

-We fully agree that direct measurements such as hepatic lipid deposition or fatty acid composition would provide more robust evidence for evaluating lipid metabolism. However, due to the limitations of our current experimental design and resources, such analyses were not included in this study. In future work, we plan to further investigate this aspect by incorporating liver histology and lipid composition analysis to gain a more comprehensive understanding of the effects of DCP on lipid metabolism. To address this, we have measured and included the triglyceride contents of liver and dorsal muscle tissues in the revised manuscript. These results provide direct evidence of lipid accumulation and further support the discussion of lipid metabolism affected by DCP inclusion (Fig. 3).

Question 4: Potential anti-nutritional factors (lines 60-78, and 374-403)

-The inroduction briely mentions anti-nutritional factors (ANFs) such as free gossypol and tannins. However, their potential residual effects on fish health and metabolism are not discussed in the results. Have the authors confirmed the absence of ANFs in the diets?

-Liver function markers (AST, ALT) fluctuated across treatments (Lines 261-270). Could this be linked to residual gossypol toxicity? A more in-depth discussion is necessary.

Answer: We sincerely thanked the reviewer’s suggestion and comments. 

-We agree that anti-nutritional factors (ANFs), such as free gossypol and tannins, may have potential effects on fish health and metabolism. We have added the content of free gossypol in DCP to the revised manuscript (Lines 110-111). Furthermore, we have expanded the discussion section to address the potential impacts of residual free gossypol on fish health and metabolism (Lines 474-476, 568-571). We appreciate the reviewer’s suggestion, and we will consider exploring this aspect in more detail in future studies.

-We have added a discussion on the potential influence of residual gossypol on AST and ALT fluctuations at higher DCP inclusion levels (Lines 291-292).

Question 5: Statistical concerns (Lines 202 206, and Tables 3-10)

-Some reported p-values and significance levels appear inconsistent (e.g., Table 3: VSI shows significant differences, but no clear trend is described). Were multiple comparison tests performed?

-Tukey's post-hoc test is mentioned, but the justification for its use over other methods (e.g., Bonferroni correction) is missing. A brief explanation of statistical choices would enhance clarity.

Answer: We sincerely thanked the reviewer’s suggestion and comments. 

-We have carefully re-examined the results presented in Table 3-10. 

-We have added a brief justification for the use of Tukey’s post-hoc test in 2.8.

Question 6: Language and readability (entire manuscript)

-There are frequent grammatical errors and awkward sentence constructions throughout the manuscript. For example, in the abstract (Lines 25-40), "Although DCP30 could lower activities of AST and ALTD, CP40 could heighten levels of LDL-C, TG, TC, ALP and BUN." The phrase is unclear and should be restructured for better readability .

-The manuscript would benefit from thorough language editing to improve readability and coherence.

Answer: We sincerely thanked the reviewer’s suggestion and comments.

-We acknowledge the grammatical issues and unclear sentence construction in the original version. we have thoroughly revised the manuscript to improve grammatical accuracy, readability, and overall coherence. The revised version has been carefully proofread to ensure clarity of expression throughout.

- We have thoroughly revised the manuscript to improve its language quality, readability, and coherence. The revised version has been carefully edited for grammar, sentence structure, and clarity. We hope the current version meets the reviewer’s expectations.

Question: 7: Title and keywords

-The title is clear, but "hormone indices" could be more specific. Consider rewording to "endocrine responses" for clarity.

-Keywords should include "anti-nutritional factors" and "feed efficiency"' to improve searchability.

Answer: We sincerely thanked the reviewer’s suggestion and comments. 

-We have revised the title to replace “hormone indices” with “endocrine responses”.

- While we agree that “anti-nutritional factors” and “feed efficiency” are important topics in aquaculture nutrition, they are not the primary focus of this study. Therefore, we have chosen not to include them as keywords in order to better reflect the central scope of the manuscript. We appreciate the reviewer’s understanding and appreciate your professional and detailed advice again.

Question 8: Figures and Tables

- Figure 1 (Digestive enzyme gene expression) and Figure 2 (Lipid metabolism gene expression) lack clear legends. Define abbreviations used in figure labels.

-Table 5 (Amino acid composition) should indicate whether differences are statistically significant.

Answer: We sincerely thanked the reviewer’s suggestion and comments. 

-We have improved the legends of Figures 1 and 2 by clearly defining all abbreviations used in the figure labels.

-The differences in Table 6 have been described in Lines 248–249 of the revised manuscript.

Question 9: References

-Several citations in the introduction (e.g., Lines 44 49) refer to older studies. Consider incorporating more recent references (post-2020) on plant protein alternatives in aquafeeds.

Answer: We sincerely thanked the reviewer’s suggestion and comments. We have updated the introduction by incorporating several recent references (published after 2020) related to plant protein alternatives in aquafeeds.

Round 2

Reviewer 1 Report

Comments and Suggestions for Authors

NO

Reviewer 3 Report

Comments and Suggestions for Authors

Dear authors,

The authors have thoroughly and thoughtfully addressed all concerns raised during the first round of peer review. The revised manuscript demonstrates substantial improvement in clarity, scientific rigor, and overall readability. Key issues related to methodology, data interpretation, anti-nutritional factors, and statistical analysis have been satisfactorily resolved. In addition, the language editing has enhanced the coherence and flow of the manuscript.

The authors have made comprehensive and effective revisions to the manuscript. All previous concerns have been addressed adequately, and no further changes are necessary. Therefore, I recommend the manuscript for acceptance in its current form.

Sincerely,